



# Synchronizing early Eocene deep-sea and continental records – new cyclostratigraphic age models from the Bighorn Basin Coring Project

Thomas Westerhold[1], Ursula Röhl[1], Roy Wilkens[2], Philip D. Gingerich[3], Will Clyde[4], Scott Wing[5], Gabe Bowen[6], Mary Kraus[7]

[1]MARUM – University of Bremen, Bremen, 28359, Germany
[2]Hawaii Institute of Geophysics & Planetology, University of Hawaii, Honolulu, HI, 96822, U.S.A.
[3]Museum of Paleontology, University of Michigan, Ann Arbor, Michigan 48109-1079, U.S.A.
[4]Department of Earth Sciences, University of New Hampshire, 56 College Rd., Durham, NH 03824,
U.S.A.
[5]Department of Paleobiology, P.O. Box 37012; National Museum of Natural History, Smithsonian
Institution, Washington, D.C. 20013 U.S.A.
[6]Department of Geology & Geophysics, University of Utah, Salt Lake City, UT, 84112, U.S.A.
[7]Department of Geological Sciences, University of Colorado at Boulder, UCB 399; Boulder, CO80309,
U.S.A.

*Correspondence to*: Thomas Westerhold (twesterhold@marum.de)

**Abstract.**

A consistent stratigraphic framework is required to understand the effect of major climate perturbations

of the geological past on both marine and terrestrial ecosystems. Transient global warming events in the

early Eocene, 56-54 Ma ago, show the impact of large scale input of carbon into the ocean-atmosphere

system. Here we provide the first time-scale synchronization of continental and marine deposits spanning

the Paleocene-Eocene Thermal Maximum (PETM) and the interval just prior to the Eocene Thermal

Maximum 2 (ETM-2). Cyclic variations in geochemical data come from continental drill cores of the

Bighorn Basin Drilling Project (BBCP, Wyoming, USA) and from marine deep-sea drilling deposits

retrieved by the Ocean Drilling Program (ODP). Both are dominated by eccentricity modulated

precession cycles that are used to construct a common cyclostratigraphic framework. Integration of age

models results in a revised astrochronology for the PETM in deep-sea records that is now generally

consistent with independent [3]He age models. The duration of the PETM is estimated at ~200 kyr for the

CIE and ~120 kyr for the pelagic clay layer. A common terrestrial and marine age model shows a

concurrent major change in marine and terrestrial biotas ~200 kyr before ETM-2. In the Bighorn Basin,

the change is referred to as Biohorizon B, and it represents a period of significant mammalian turnover

and immigration, separating the upper *Haplomylus-Ectocion* Range Zone from the *Bunophorus* Interval

Zone and approximating the Wa-4–Wa-5 land mammal zone boundary. In sediments from ODP Site 1262

(Walvis Ridge), major changes in the biota at this time are documented by the radiation of a "2nd

generation" of apical spine-bearing sphenoliths species (e.g., *S. radians* and *S. editus*), the emergence of

*T. orthostylus*, and the marked decline of *D. multiradiatus*.



## 1 Introduction

Early Eocene greenhouse climate on Earth was punctuated by transient global warming events (Cramer et al., 2003; Kirtland-Turner et al., 2014). The Paleocene-Eocene Thermal Maximum (PETM or Eocene Thermal Maximum 1: ETM1) at 55.93 Ma (Kennett and Stott 1991; Koch et al., 1992; Bowen et al., 2001;

Zachos et al., 2005; Bowen et al., 2015) is the most pronounced hyperthermal event, with a ~5-8°C global warming (McInerney and Wing, 2011). The Elmo event (ETM2 or H1) (Cramer et al., 2003; Lourens et al., 2005) is another major transient warming event at 54.05 Ma (Westerhold et al. 2017). Both have been studied in great detail in both in deep-sea sedimentary and terrestrial successions (Zachos et al., 2005; Abels et al., 2016).

The hyperthermal events in outcrops and ocean drill cores can be identified by the characteristic negative carbon isotope excursions (CIEs), although these differ in magnitude (McInerney and Wing, 2011; Bowen, 2013). The CIEs are interpreted as massive inputs of $\delta^{13}$C-depleted carbon to the exogenic carbon pool (see Dickens et al., 2011 for discussion). For both events, major changes in land and ocean biotas have been documented (McInerny and Wing, 2011; Slujis et al., 2007). In the fossil record, the PETM,

for example, marks the appearance of the modern order of mammals including horses and primates on land (Gingerich, 1989; Gingerich, 2006) and a major extinction of benthic foraminifera in the deep sea (Thomas, 1989).

The hyperthermals provide important evidence for understanding the dramatic and long-lasting consequences of a rapid massive input of $CO_2$ into the ocean-atmosphere system within a few thousand

years (Kirtland-Turner and Ridgwell, 2016; Zeebe et al., 2016). The PETM in particular, the largest CIE of the last 100 million years, is very important for understanding the future of climate on Earth (McInerney and Wing, 2011), but the anthropogenic input of $CO_2$ may be as much as an order of magnitude more rapid than at the PETM (Zeebe et al., 2016).

Key records for studying the early Eocene climate and hyperthermals come from carbonate rich deep-sea

drill cores from Walvis Ridge in the South Atlantic (Zachos et al., 2005) and terrestrial fluvial deposits with paleosols in the Bighorn Basin in Wyoming, USA (Koch et al., 1992; Bowen et al., 2001, 2015; Abels et al., 2012; Abels et al., 2016). Deep-sea records have a much lower sedimentation rate (cm/kyr) compared to the terrestrial records (m/kyr), but have been deposited continuously. Sedimentation at the terrestrial successions very likely was more dynamic due to the different types of deposition (see Bowen

et al., 2015). To interpret rates of changes and processes before, during and after the events a detailed age model is required. Deep-sea records around the PETM and Elmo event reveal extraordinary cyclicity related to precession and eccentricity of the Earth's orbit that was used for establishing high-resolution age models based on cyclostratigraphy and astronomical tuning (Lourens et al., 2005; Röhl et al., 2000, 2007; Westerhold et al., 2007; Abdul Aziz et al., 2008; Stap et al., 2009). Estimates for the duration of

the PETM from deep-sea records are complicated by severe carbonate dissolution, which forms a clay-rich layer at the onset of the event (Röhl et al., 2007). Alternative age models based on extraterrestrial



[3]He assign more time to the clay layer and a more rapid recovery to pre-PETM $\delta^{13}C$ values than does the cyclostratigraphic model (Murphy et al., 2010). The stacking pattern of paleosols in the terrestrial PETM section at Polecat Bench is driven by climatic precession and therefore was used to develop an astronomical age model for the CIE (Abdul-Aziz et al., 2008). Cyclicity in outcrops of fluvial deposits

prior to and during the Elmo event in the Bighorn Basin was dominated by precession (Abels et al., 2012; Abels et al., 2013), similar to what has been found in deep-sea sediments (Lourens et al., 2005; Westerhold et al., 2007; Zachos et al., 2010; Littler et al., 2014).

Because records from both realms show precession-dominated cyclicity it should be possible to correlate and synchronize them. In summer 2011, the Bighorn Basin Coring Project (BBCP) drilled 900 m of

overlapping cores from three sites covering the interval of the PETM and Elmo events (Clyde et al., 2013). The BBCP retrieved continuous unweathered material for common multiproxy studies. The purpose of this report is to establish high-resolution age models for the BBCP drill cores based on cyclostratigraphy and integrate existing age models from outcrops. Second, these new BBCP drill cores age models will be combined with deep-sea records to synchronize and improve the available astronomical age model for

the PETM and Elmo interval. The new age models will allow other studies to compare multiple proxy records from both realms at unprecedented temporal accuracy. Our study also allows us to consider whether the mammalian turnover called "Biohorizon B" in the Bighorn Basin fossil faunas (Schlanker, 1980; Chew and Oheim, 2009) is synchronous with changes in deep-sea calcareous nannofossil assemblages (Agnini et al., 2007) significantly prior to the Elmo event.

**2 Material and Methods**

The BBCP drilled late Paleocene to early Eocene fluvial deposits including the PETM at Polecat Bench (PCB) and Basin Substation (BSN), and the Elmo interval at Gilmore Hill (GMH) (Fig.1, Clyde et al., 2013). Two overlapping holes were drilled at BSN down to 138.4 mbs (BSN11-1A) and 138.6 mbs (BSN11-1B). At PCB two overlapping holes were drilled to 130 mbs (PCB11-2A) and 245.1 mbs

(PCB11-2B). At GMH one hole was drilled down to 202.4 mbs (GMH11-3A) and a second down to 66.7 mbs (GMH11-3B). All cores were split and processed according to IODP standards that included visual core description, color and line-scanning, sampling for post party investigations at home laboratories, and archiving at the Bremen Core Repository during a BBCP Science Party at MARUM, University Bremen, Germany, in January 2012 (Clyde et al., 2013).

Here we present the results of processing and interpreting line scan images and color reflectance data for the BSN and GMH and PCB sites, with the records from PCB being presented initially in Bowen et al. (2015). All cores from PCB, BSN and GMH were XRF scanned over the course of 2012 at MARUM, University of Bremen, and we will make use of iron (Fe) intensity data here as well. XRF data were collected every 2 cm down-core using XRF core scanner 3 (AVAATECH Serial No. 12) at MARUM –





University Bremen, over a 1.2 cm$^2$ area with a down-core split size of 10 mm using a generator settings of 50, 30 and 10 kV, a respective current of 1.0, 1.0 and 0.2 mA, and a sampling time of 10 seconds in each run directly at the split core surface of the archive half. The split core surface was covered with a 4-micron thin SPEXCerti Perp Ultralene1 foil to avoid contamination of the XRF detector-prism and

desiccation of the cores. The data were acquired by a Canberra X-PIPS Silicon Drift Detector (SDD; Model SXD 15C-150-500) with 150eV X-ray resolution, the Canberra Digital Spectrum Analyzer DAS 1000 and an Oxford Instruments 100W Neptune X-ray tube with a Rhodium (Rh) target. Raw data spectra were processed by the Analysis of X-ray spectra by Iterative Least square software (WIN AXIL) package from Canberra Eurisys. Core data and images have been correlated and integrated using the new software

tool CODD (Code for Ocean Drilling Data, Wilkens et al. 2017). This tool greatly facilities handling of large and complex data sets and allows to use core images for scientific analysis.

Pedogenic carbonate nodules for isotope analysis were identified as discrete, small (~2mm to >5cm in diameter), well-cemented, rounded to sub-rounded accumulations of micritic carbonate, some of which contain observable secondary, diagenetic spar. Samples larger than ~1 cm diameter were slabbed to

expose a clean flat surface. For smaller samples, the exterior surface of the nodule was etched using a dental drill. Primary micrite was collected with a dental drill under a binocular microscope. Aliquots of approximately 100 μg were weighed into 10 ml (for analyses conducted at the University of Utah) or 4 ml (Purdue University) Exitainer vials. The samples were reacted with orthophosphoric acid at 75° C for a minimum of one hour and analyzed using a ThermoFinnigan Gas Bench II coupled to a Delta Plus

(University of Utah SIRFER lab) or Delta V (Purdue Stable Isotope facility) Isotope Ratio Mass Spectrometer. Analyses of reference carbonates (Carrara Marble, NBS-19 and LSVEC) were used to correct the measured sample values to the Vienna PeeDee Belemnite (VPDB) scale. Analytical precision was approximately 0.1‰, for both $\delta^{13}$C and $\delta^{18}$O, at both labs based on replicate analyses of reference carbonates throughout the course of the analyses. Repeatability of the pedogenic carbonate analyses

averaged 0.20‰ for $\delta^{13}$C and 0.24‰ for $\delta^{18}$O based on replicate analyses of pedogenic carbonate samples. Isotopic data for PCB nodules were previously reported in Bowen et al. (2015).

All data and tables from this study will be open access available at Pangaea Database (www.pangaea.de) upon acceptance of the manuscript, for reviewers a dataset file is available from the editor in the discussion phase.

**3 Results**

**3.1 Color reflectance data, XRF core scanning data and composite depth**

Weathering resulted in brighter, yellowish colors within the upper ~30 meters of the BBCP drill cores (Fig. 2 in Clyde et al., 2013). Below the weathering zone sediments appear light to dark gray with reddish to purplish colored paleosols. Lightness varies between 20-60% for all sites with a high degree of

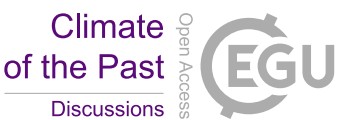

variability. a* (red/green) values for PCB and BSN cores range from 0 with values up to 10 or higher in reddish and purplish paleosols. a* values of the GMH cores show higher variability due to the more reddish, varying lithology. b* (blue/yellow) values vary more in the upper 30 meters of the cores in line with the more yellow colors of the weathered zone. Fe core scanning intensities vary between several

1000 up to 100000 total counts (area) and display cyclic changes with a much better signal to noise ratio than the color data. BSN Fe intensities reveal lower variability then PCB and GMH, but higher Fe peaks are more common. GMH and PCB Fe data show persistent cyclicity throughout the succession.

Core images, color reflectance and Fe data have been utilized to correlate between parallel holes for PCB, BSN, and GMH drill cores. As normal shipboard routine for multiple holes drilled by IODP, cores of the

BBCP were offset from the original drilling depth (mbs) and subsequently combined. The correlation of the two PCB records and the resulting composite can be found in Bowen et al., (2015, their Fig. 1). Lithological logs, core images, color reflectance CIE L*a*b* from the BBCP Science Party (Clyde et al., 2013), and XRF core scanning Fe intensity for BSN, GMH, and PCB are provided in supplementary Figures S1, S2 and S3 and data tables S1-S6.

**3.2 Timeseries analysis of BBCP drillcores**

Evolutionary Power Spectra were calculated for the XRF score scanning Fe intensities and color reflectance a* (AStar) data for all drill holes separately. Before spectral analysis, the raw data were re-sampled and the trend removed. On the residual data Multitaper Method (MTM) spectra were calculated using the SSA-Toolkit (Cite) using 3 tapers and resolution of 2 (Ghil et al., 2002), with confidence levels

based on a robust red noise estimation (Mann and Lees, 1996). Cycles identified by MTM spectrum analysis were then extracted by a Gaussian bandpass filter of the central frequency with 30% bandwidth using AnalySeries (Paillard et al., 1996).

**3.2.1 Polecat Bench**

At PCB, the dominant cycles in Fe intensity data are 8.2, 3.45, 1.2, 1.02 and 0.58 m long (Fig. 2), and in

the a* data 7.8-8, 3.45 and 1.1m (Fig. S6). This result is consistent with the observation of 7.7, 3.3 and 1.1 m long cycles in a* data obtained from the outcrop (Abdul-Aziz et al., 2008). The slight difference in length could be due to thickness variations of paleosols and sandstone beds between the drill site location and the outcrop. The two longer cycles around 8 and 3.5m have been interpreted as precession and half precession cycles also present in Plio-Pleistocene successions (see Abdul-Aziz et al., 2008). Assuming

that the 8.2 m cycles represent the averaged 21 kyr long precession cycle, the other cycles have a period of 8.83, 3.1, 2.61 and 1.48 kyrs, respectively. The evolutionary spectra show some slight changes in the length of the dominant cycles around 8 m which can be interpreted as changes in sedimentation rate.



### 3.2.2 Basin Substation

At BSN, a range of dominant cycles in Fe intensity are apparent (6.8, 3.3, 2, 1.7, 1.25, 1.05 and 0.58 m, Fig. 3). In the a* data, a similar variety can be observed (Fig. S7). The evolutionary spectra shows that cycles are not persistent over the entire succession, hampering the construction of a cyclostratigraphic age model at BSN. These irregularities in cyclicity could point to strong changes in sedimentation rate, condensed intervals in the record, and/or changes in the processes regulating the Fe content in the sediment. Relatively stable cycles can be observed in the interval from 50 to 85 m, with dominant 6.8 m cycles that might correspond to the precession cycles with ~8 m length in the PCB core. If so, the 3.3 m cycles could represent the half-precession cycles. Because the cycles are not persistent for the entire drill core, a cyclostratigraphy for BSN was not established here.

### 3.2.3 Gilmore Hill

At GMH the dominant cycles in Fe intensity and a* data are 7.5-6.7, 3.3, 2.5, ~1m and 0.62 m long (Fig. 2, Fig. S7). The evolutionary spectra show relatively regular cycles just before the Elmo event (see discussion below), which is cut out by sandstones of a channel fill at 0-20 m. Below this, the dominant cycle undulates around 7 m and is clearly visible in the data. Long-term sediment accumulation rates in the closest outcrop section with magnetostratigraphy (Clyde et al., 1994; Clyde, 2001) suggest that 7 meters of sediment in this part of the basin represents between 14,500 and 25,500 years and thus is consistent with precession. The 7 m and 3.3 m long cycles are also consistent with precession and half-precession cycles observed in a* values from the Bighorn Basin Deer Creek Amphitheater section, which is ~17km away along the McCullough Peaks escarpment (Abels et al., 2013). Thus these cycles can be used for building a cyclostratigraphy.

## 4 Cyclostratigraphy - Linking outcrops and BBCP drillcores

The obtained data allow the construction of a cyclostratigraphy for drill cores from GMH and PCB. We correlated GMH and PCB records with outcrop successions to allow full integration of Bighorn Basin surface-derived data with the drill cores.

### 4.1 Age model for PCB PETM

For the PCB drill cores, a correlation to a composite outcrop (Abdul Aziz et al., 2008) is already available (Bowen et al., 2015) identifying well-known local marker beds consisting of red to purplish paleosols. The rhythmic stacking pattern of the paleosols is driven by orbital precession cycles and was successfully used to establish a cyclostratigraphy for the PETM at Polecat Bench (Abdul Aziz et al., 2008) roughly consistent with independent age models from deep sea cores (Farley and Eltgroth, 2003; Röhl et al., 2007). For the PCB cores a set of pedogenic age models was developed calculating time based on thickness,





sediment type, and paleosol maturity of individual stratigraphic units (Bowen et al., 2015), but no cyclostratigraphy similar to the outcrop has been developed so far.

The cyclostratigraphy for PCB cores presented here is based on cycle counting of both the a* and XRF Fe data cycles that are interpreted to be related to precession. Precession and half precession cyclicity was

extracted by Gaussian filtering of a* and XRF Fe data (Figure 5) with the Fe data providing a much cleaner rhythmic signal due to the better signal to noise ratio compared to a*. Taking the soil nodule carbon isotope data (Bowen et al., 2015) as reference, we started counting at the onset of the PETM CIE (see Figure 5), with positive numbers up-core and negative numbers down-core. The filter of the precession cycles of ~8.2 m in both data show modulations that are consistent with eccentricity. We

estimate that the PCB core covers 33 precession cycles, of which we use 29 for a direct age assignment. Cycles -10, -11, -13 and -14 are hard to identify because of the low amplitude variation in the data. This observation is consistent with lower amplitudes in a* and XRF Fe data 300 kyr prior to the onset of the PETM in deep-sea sediments from Walvis Ridge (South Atlantic) and Blake Nose (North Atlantic) (Westerhold et al., 2007) caused by a minimum in the long 405-kyr eccentricity cycle.

For each of the precession cycles we assume a constant duration of 21 kyr as done in previous deep-sea cyclostratigraphic models (Röhl et al., 2007; Westerhold et al., 2007) setting the onset of the PETM as 0. Relative ages with respect to the onset of the PETM are given in table 1. For absolute ages we use the age for onset of the PETM as 55.930 Ma (Westerhold et al., 2007; Westerhold et al., 2015) and add or subtract the relative age. According to the filter of the a* and Fe data, we also assume that the onset of the PETM

is located in the minimum between cycle -1 and 1 (Figure 5). The new cyclostratigraphy is almost identical to the outcrop model (Abdul-Aziz et al., 2008) for the PETM. Because we use the longer drill core from PCB now as a reference, minor discrepancies between outcrop and drill core paleosol and sandstone bed thicknesses had to be corrected in the tie points for the outcrop (Abdul-Aziz et al., 2008). This did not affect the original estimates for the duration of the PETM reported in Abdul-Aziz et al.

(2008). The new cyclostratigraphic age model for PCB covers the latest Paleocene from precession-cycle -15 (as in Westerhold et al., 2007) to early Eocene precession-cycle 15, this is equivalent to the 56.234 to 55.626 Ma interval with the PETM at 55.930 Ma.

## 4.2 Age model for GMH – pre Elmo interval

Extensive outcrop work and cyclostratigraphic interpretations are also available for terrestrial sediments

across the ETM-2 (Elmo, Lourens et al., 2005) in the Bighorn Basin (Clyde et al., 1994; Clyde et al., 2007; Abels et al., 2012; Abels et al., 2013; Abels et al., 2016; D'Ambrosia et al., 2017). Alluvial sedimentary cycles before and after the Elmo are shown to be precession forced (Abels et al., 2012). The GMH cores cover an interval prior to the ETM-2, which is cut out by a sandstone channel complex (Clyde et al. 2013). To establish a cyclostratigraphy for the GMH drill cores, precession and half-precession

cycles were extracted from the a* and XRF Fe data (Figure 6). Both data show high amplitude variations



as expected from the more reddish terrestrial deposits. The extracted a* and XRF Fe data cycles are consistent with previous cyclic variations in a* values from outcrop samples. The drill core data have been correlated to the Deer Creek (Fig. S9, Abels et al., 2012) and Gilmore Hill (Fig. S10, D'Ambrosia et al., 2017) sections using prominent purple marker beds and field observations. Subsequently, we

adopted the labelling system of Abels et al. (2013, A-P) but extend it towards the ETM-2 at GMH (Q to Y). As already observed in the PCB drill cores, data from the GMH core show the same precession length and amplitude modulation as observed in outcrop samples (Fig. S9). Combining the correlation to the outcrop interpretation with the precession-filtered data from the drill cores we established a best fit cycle counting age model for GMH drill cores. This is necessary because the filter of a* and Fe data are not

consistent in all parts of the record. Basic tie-points for correlation and cyclostratigraphy are given in table 2. For a relative age model we assigned the precession cycle numbers identified in Walvis Ridge ODP sites from Leg 208 and their relative age to the onset of the PETM (Westerhold et al., 2007) to the precession cycles found in GMH. For absolute ages we provide one option which refers to the age of 55.930 Ma for the onset of the PETM (see PCB) and another option relative to the age of 54.05 Ma for

ETM-2. The age model suggests that the GMH cores cover precession cycles -A to Y, representing roughly 500 kyrs of terrestrial deposition from 54.596 to 54.071 Ma.

**5 Discussion**

The objective of this study is not only to provide a high resolution cyclostratigraphic age model for the BBCP drill cores but also to improve existing age models for the interval spanning the PETM and prior

to ETM-2. The BBCP drill core and Bighorn Basin outcrops are an ideal basis for revision of the PETM age model, which is mainly derived from deep-sea sediments with low sedimentation rates. Dissolution of carbonate, particularly at the onset of the PETM, hampers the establishment of a complete chronology for the PETM from marine cores (Kelly et al., 2005; Kelly et al., 2010). Uncertainty in correlation between terrestrial successions from the Bighorn Basin and marine records also makes it difficult to evaluate if the

terrestrial Biohorizon B (Schlanker, 1980; Chew and Oheim, 2009) is synchronous with major changes in marine calcareous nannofossil assemblages prior to ETM-2 (Agnini et al., 2007). The extremely high sedimentation rates of the Bighorn Basin deposits of the Fort Union and Willwood formations allow very detailed insight to changes in biota on land that can be synchronised with the marine records by using a common astrochronology.

**5.1 Synchronizing deep-sea and continental records for the PETM**

Precession related variations in XRF core scanning data (Röhl et al., 2000; Röhl et al., 2007) on the one hand, and the concentration of extraterrestrial [3]He in ODP cores from the same regions (Farley and Eltgroth 2003; Murphy et al., 2010) were used to develop an age model for the PETM from pelagic sequences. During the PETM, massive dissolution of carbonates in the deep sea truncated the record





(Zachos et al., 2005), complicating the age model constructions. Cyclostratigraphy and extraterrestrial $^3$He age models are roughly consistent but show some discrepancies in the duration and recovery of the CIE (Murphy et al., 2010). The extraterrestrial $^3$He age model proposed $217^{+44/-33}$ kyr for the entire CIE compared to 171 kyr by cyclostratigraphy (Röhl et al., 2007; Murphy et al., 2010). The duration of the

clay layer, which is the interval between the sharp contact of dissolution at the onset and the recovery of carbonate content to pre-event levels (Figure 7), was estimated to be $167^{+43/-24}$ kyr by extraterrestrial $^3$He (Murphy et al., 2010) and ~95 kyr by cyclostratigraphy (Röhl et al., 2007). The helium age model can be compromised if the flux of extraterrestrial $^3$He was not constant during the PETM. Orbital chronology depends on the correct recognition of all sedimentary cycles, which is notoriously difficult at the onset of

the PETM. Severe carbonate dissolution, including the burn-down of carbonate deposited before the actual ocean acidification during the onset of the PETM (Zeebe and Zachos, 2007), could explain the difference in these estimates of the PETM duration. In addition, cyclostratigraphic work around the deep-sea PETM sections (Röhl et al., 2007; Westerhold et al., 2007) limits the maximum duration of the clay layer to 7 precession cycles, or 147 kyr, which is within the error of the $^3$He age model (210-143 kyr,

Murphy et al., 2010).

     At the Walvis Ridge ODP sites, the top of the clay layer coincides with the top of the initial rapid recovery of the CIE (Recovery phase I in Murphy et al., 2010). To correlate deep-sea and terrestrial records, the onset and the top of the initial rapid recovery of the CIE are commonly used (McInerney and Wing, 2011). The PCB cyclostratigraphy indicates that the duration of this interval mentioned above covers six

precession cycles or ~120 kyr (assuming an average duration of 21 kyr for one precession cycle). This duration is between the estimates of 90 and 135 kyr used as a basis for the age model in Bowen et al. (2015). As a consequence, a duration of 120 kyr suggests that the deep-sea records (~95 kyr) are missing ~25 kyr or about one precession cycle at the onset of the event. Because the onset of the PETM was set into a precession minimum in the PCB cyclostratigraphy model (see above), the additional 4 kyr on top

of the missing precession cycle in the duration compared to the deep sea can be explained. If we compare the PCB CIE of our new cyclostratigraphy to the deep-sea CIE using the existing orbital chronology (Röhl et al., 2007) and aligning the records at the onset of the PETM, the initial recovery from sustained minimum $\delta^{13}$C values will be offset by ~25 kyr. A lag of 25kyr between this major inflection in the ocean and atmosphere carbon isotope records is not possible given the rapid rate of carbon exchange between

the atmosphere and surface (10's of years) and deep (100's of years) ocean reservoirs (Revelle and Suess, 1957; Broecker and Peng, 1982; Bowen, 2013). We therefore modified and updated the age model of Röhl et al., (2007) from precession cycles -20 to 20 (56.320 to 55.520 Ma) by adding a precession cycle at the onset of the PETM and moving the onset of the PETM between two precession cycles (see Table 1, Fig. 7 and S12).

As a result of the updated orbital chronology, the duration of the clay layer now is estimated to be ~120 kyr, and the total duration of the PETM is ~200 kyr, compared to the previous 95 and 171 kyr estimates





(Röhl et al., 2007). The updated duration of the CIE using cyclostratigraphy is in agreement now with the [3]He age model (Fraley and Eltgroth, 2003; Murphy et al., 2010), but the duration of the clay layer is still more than 23 kyr shorter. Although the overall cyclostratigraphy around the PETM (Röhl et al., 2007; Westerhold et al., 2007) would allow a seventh precession cycle in the clay layer, there is no indication

in the Bighorn Basin records for a missing precession cycle at the onset of the event. Six cycles in the PETM CIE have also been reported from the BH9/05 drill core of the Paleocene-Eocene boundary in Spitsbergen (Charles et al., 2011; cyclostratigraphic option B therein). U/Pb dating for the onset of the PETM from the Spitsbergen record (55.866 ± 0.098 Ma) is also consistent with the 55.930 ± 0.05 Ma estimate from astronomical calibration (Westerhold et al., 2008; Westerhold et al., 2015).

Direct comparison of the [3]He age model (Murphy et al., 2010) and our orbital chronology for Site 1266 (Fig. S13) shows that the initial rapid recovery located in precession cycle 6 of the PCB record is, on average, offset by 40 kyr in the [3]He age model. For calibration of the constant extraterrestrial [3]He flux, a sedimentation rate between 306.92 and 308.54 mcd at Site 1266 of 1.43 cm/kyr on average based on the age model of Röhl et al. (2007) was used. It is important to note that the age model uses the average

duration for precession of 21 kyr. Over hundreds of millions of years, the Moon moved away from Earth and the rotation of the Earth slowed down with time, changing the precession frequency of Earth (Laskar et al., 2004). As a consequence, the average precession cycle at 56 Ma will not be 21kyr but rather 20.5 kyr. This subtle difference could point to the fact that, in the calibration interval of the 6 precession cycles used for the extraterrestrial [3]He flux estimate, there is less time and thus the sedimentation rates are a bit

higher, on the order of 1.5 cm/kyr. Assuming the decrease in the precession period over time to be valid, the extraterrestrial [3]He flux would be higher, resulting in a duration of ~200 kyr for the CIE (306.15 to 304.70 mcd at 1266) and ~160 kyr for the clay layer. The discrepancy would be still hard to explain because to match the duration of the clay layer to the ~120 kyr estimate, the sedimentation rate in the calibration interval has to be increased to 2 cm /kyr resulting in a cycle duration of 14-16 kyr for the six

precession cycles. The duration of the CIE would also be reduced to 150-160 kyr, making this scenario rather unlikely. These uncertainties call for more [3]He-based studies across the PETM to investigate if there is also time missing in the PCB sections. Until then we consider the updated age model presented here as a proper solution to compare deep sea with terrestrial records.

## 5.2 Synchronizing deep-sea and continental records prior to the ETM-2

Eccentricity modulated precession cycles dominate geochemical records in the interval prior to ETM-2 and have been used to establish cyclostratigraphic age models (Westerhold et al., 2007; Zachos et al., 2010; Littler et al., 2014). To synchronize marine and terrestrial records (GMH), we simply adopted the cyclostratigraphic ages from the deep-sea sections (Fig. 8, Tab. 2). Remarkably, the same eccentricity related amplitude modulation of XRF Fe data can be observed for both records. Low amplitude precession

cycles are present from 54.65 to 54.48 Ma followed by high amplitude cyclicity around 54.44 Ma,



implying a climate system feedback to modulations in precession affecting both realms. Variations in the XRF Fe data of the deep sea are most likely driven by changes in carbonate deposition. In the terrestrial sediments these variations have been attributed to large-scale reorganization of the fluvial system driven by astronomically forced changes in the hydrological cycle (Abels et al., 2012). Comparing the stable

carbon isotope curves from deep-sea benthic foraminifera, bulk carbonate sediment and soil nodules (Tab. S9) shows similar congruent variations, even outside the extraordinary hyperthermal events (e.g. ETM-2, Abels et al., 2016). Consistent small scale variability clearly is linked to changes in the global carbon cycle.

Knowing that the age model for the deep-sea and the Bighon Basin records are synchronous, we can test

for a temporal relation between Biohorizon B in the Bighorn Basin (Clyde et al., 2007) and biotic changes in calcareous nannofossils in deep-sea records. In sediments from ODP Site 1262 (Walvis Ridge), major changes in the biota are documented by the radiation of a "2nd generation" of apical spine-bearing sphenoliths species (e.g., *S. radians* and *S. editus*), the emergence of *T. orthostylus*, and the marked decline of *D. multiradiatus* 200 kyr prior to the ETM-2 (Agnini et al., 2007; Figure 8). Based on our

GMH cyclostratigraphy, the faunal turnover at Biohorizon B occurred almost at the same time, pointing to a common response of the biota to environmental change 54.25 Ma ago. Biohorizon B marks a major period of mammalian turnover and immigration, separating the upper Haplomylus-Ectocion Range Zone from the Bunophorus Interval Zone and approximating the Wa-4–Wa-5 zone boundary (Schankler, 1980; Chew, 2009; Chew and Oheim, 2013; Chew, 2015, Clyde et al., 2007). In high-resolution isotope records

from the deep sea (Littler et al., 2014), there is no evidence for a major perturbation or even change in environmental conditions that could cause the synchronous response in the marine and terrestrial ecosystems, but the coincident timing could point to some common response to a forcing yet to be discovered.

## 6 Conclusions

Fe intensities, core images, and color reflectance data were used to build composite records for the Bighorn Basin Coring Project drill cores from Gilmore Hill, Basin Substation, and Polecat Bench. Eccentricity modulated precession scale cyclicity observed in the high-resolution data allowed the construction of cyclostratigraphic age models for GMH and PCB spanning a 500 kyr interval prior to the ETM-2 and a 500 kyr interval across the PETM. The established orbital chronology of the drill core data

not only is consistent with previous age models from outcrops, but also helped to improve the cyclostratigraphic age model for the PETM in deep-sea records. Synchronisation and integration of all records define a duration of ~200 kyr for the CIE and ~120 kyr for the clay layer of the PETM, largely consistent with independent $^3$He age models. The combination of marine and terrestrial records on a common age model prior to the ETM-2 shows a almost synchronous change in marine and terrestrial

biota that is likely related to a common but as yet unknown environmental change. The successful




Bowen, G. J.: Up in smoke: A role for organic carbon feedbacks in Paleogene hyperthermals, Global and Planetary Change, 109, 18-29, http://dx.doi.org/10.1016/j.gloplacha.2013.07.001, 2013.

Bowen, G. J., Maibauer, B. J., Kraus, M. J., Röhl, U., Westerhold, T., Steimke, A., Gingerich, P. D., Wing, S. L., and Clyde, W. C.: Two massive, rapid releases of carbon during the onset of the Palaeocene-Eocene thermal maximum, Nature Geosci, 8, 44-47, 10.1038/ngeo2316, 2015.

Broecker, W.S., Peng, T.-H.: Tracers in the sea. Lamont-Doherty Geological Observatory, 690 pp., 1982.

Charles, A. J., Condon, D. J., Harding, I. C., Pälike, H., Marshall, J. E. A., Cui, Y., Kump, L., and Croudace, I. W.: Constraints on the numerical age of the Paleocene-Eocene boundary, Geochem. Geophys. Geosyst., 12, Q0AA17, 10.1029/2010gc003426, 2011.

Chew, A.E.: Paleoecology of the early Eocene Willwood mammal fauna from the central Bighorn Basin, Wyoming. Paleobiology 35, 13–31, 2009.

Chew, A. E., and Oheim, K. B.: Diversity and climate change in the middle-late Wasatchian (early Eocene) Willwood Formation, central Bighorn Basin, Wyoming, Palaeogeography, Palaeoclimatology, Palaeoecology, 369, 67-78, http://dx.doi.org/10.1016/j.palaeo.2012.10.004, 2013.

Chew, A. E.: Mammal faunal change in the zone of the Paleogene hyperthermals ETM2 and H2, Clim. Past, 11, 1223-1237, 10.5194/cp-11-1223-2015, 2015.

Clyde, W. C., Stamatakos, J., and Gingerich, P. D. : Chronology of the Wasatchian land-mammal age: magnetostratigraphic results from the McCullough Peaks section, northern Bighorn Basin, Wyoming. Journal of Geology 102:367-377, 1994.

Clyde, W. C., Hamzi, W., Finarelli, J. A., Wing, S. L., Schankler, D., and Chew, A.: Basin-wide magnetostratigraphic framework for the Bighorn Basin, Wyoming, GSA Bulletin, 119, 848-859, 2007.

Clyde, W. C., Gingerich, P. D., Wing, S. L., Röhl, U., Westerhold, T., Bowen, G., Johnson, K., Baczynski, A. A., Diefendorf, A., McInerney, F., Schnurrenberger, D., Noren, A., Brady, K., and the, B. S. T.: Bighorn Basin Coring Project (BBCP): a continental perspective on early Paleogene hyperthermals, Sci. Dril., 16, 21-31, 10.5194/sd-16-21-2013, 2013.

Cramer, B. S., Wright, J. D., Kent, D. V., and Aubry, M.-P.: Orbital climate forcing of d13C excursions in the late Paleocene - Eocene (chrons C24n-C25n), Paleoceanography, 18, 1097, 10.1029/2003PA000909, 2003.

Dickens, G. R.: Down the Rabbit Hole: toward appropriate discussion of methane release from gas hydrate systems during the Paleocene-Eocene thermal maximum and other past hyperthermal events, Clim. Past, 7, 831-846, 10.5194/cp-7-831-2011, 2011.

D'Ambrosia, A. R., Clyde, W. C., Fricke, H. C., Gingerich, P. D., and Abels, H. A.: Repetitive mammalian dwarfing during ancient greenhouse warming events, Science Advances, 3, 10.1126/sciadv.1601430, 2017.

Farley, K. A., and Eltgroth, S. F.: An alternative age model for the Paleocene-Eocene thermal maximum using extraterrestrial 3He, Earth and Planetary Science Letters, 208, 135-148, 2003.




Gingerich, P. D.: New earliest Wasatchian mammalian fauna from the Eocene of northwestern Wyoming: composition and diversity in a rarely sampled high-floodplain assemblage. University of Michigan Papers on Paleontology, 28, 1-97. http://hdl.handle.net/2027.42/48628, 1989.

Gingerich, P. D.: Environment and evolution through the Paleocene-Eocene thermal maximum, Trends in Ecology & Evolution, 21, 246-253, 2006.

Kelly, D. C., Zachos, J. C., Bralower, T. J., and Schellenberg, S. A.: Enhanced terrestrial weathering/runoff and surface ocean carbonate production during the recovery stages of the Paleocene-Eocene thermal maximum, Paleoceanography, 20, 10.1029/2005PA001163, 2005.

Kelly, D. C., Nielsen, T. M. J., McCarren, H. K., Zachos, J. C., and Röhl, U.: Spatiotemporal patterns of carbonate sedimentation in the South Atlantic: Implications for carbon cycling during the Paleocene-Eocene thermal maximum, Palaeogeography, Palaeoclimatology, Palaeoecology, 293, 30-40, 10.1016/j.palaeo.2010.04.027, 2010.

Kennett, J. P., and Stott, L. D.: Abrupt deep sea warming, paleoceanographic changes and benthic extinctions at the end of the Paleocene, Nature, 353, 225-229, 1991.

Kirtland Turner, S., Sexton, P. F., Charles, C. D., and Norris, R. D.: Persistence of carbon release events through the peak of early Eocene global warmth, Nature Geosci, 7, 10.1038/ngeo2240, 2014.

Kirtland Turner, S., and Ridgwell, A.: Development of a novel empirical framework for interpreting geological carbon isotope excursions, with implications for the rate of carbon injection across the PETM, Earth and Planetary Science Letters, 435, 1-13, 10.1016/j.epsl.2015.11.027, 2016.

Koch, P. L., Zachos, J. C., and Gingerich, P.: Correlation between isotope records in marine and continental carbon reservoirs near the Paleocene/Eocene boundary, Nature, 358, 319-322, 1992.

Laskar, J., Robutel, P., Joutel, F., Gastineau, M., Correia, A., and Levrard, B.: A long-term numerical solution for the insolation quantities of the Earth, Astronomy and Astrophysics, 428, 261-285, 10.1051/0004-6361:20041335, 2004.

Lauretano, V., Littler, K., Polling, M., Zachos, J. C., and Lourens, L. J.: Frequency, magnitude and character of hyperthermal events at the onset of the Early Eocene Climatic Optimum, Clim. Past, 11, 1313-1324, 10.5194/cp-11-1313-2015, 2015.

Littler, K., Röhl, U., Westerhold, T., and Zachos, J. C.: A high-resolution benthic stable-isotope record for the South Atlantic: Implications for orbital-scale changes in Late Paleocene–Early Eocene climate and carbon cycling, Earth and Planetary Science Letters, 401, 18-30, 10.1016/j.epsl.2014.05.054, 2014.

Lourens, L. J., Sluijs, A., Kroon, D., Zachos, J. C., Thomas, E., Röhl, U., Bowles, J., and Raffi, I.: Astronomical pacing of late Palaeocene to early Eocene global warming events, Nature, 435, 1083-1087, 10.1038/nature03814, 2005.



McInerney, F. A., and Wing, S. L.: The Paleocene-Eocene Thermal Maximum: A Perturbation of Carbon Cycle, Climate, and Biosphere with Implications for the Future, Annual Review of Earth and Planetary Sciences, 39, 489-516, doi:10.1146/annurev-earth-040610-133431, 2011.

Murphy, B. H., Farley, K. A., and Zachos, J. C.: An extraterrestrial 3He-based timescale for the Paleocene-Eocene thermal maximum (PETM) from Walvis Ridge, IODP Site 1266, Geochimica et Cosmochimica Acta, 74, 5098-5108, 10.1016/j.gca.2010.03.039, 2010.

Paillard, D., Labeyrie, L., and Yiou, P.: Macintosh program performs time-series analysis., Eos Trans. AGU, 77, (http://www.agu.org/eos_elec/96097e.html), 1996.

Revelle, R., Suess, H.E.: Carbon dioxide exchange between atmosphere and ocean and the question of an increase of atmospheric CO2 during the past decades. Tellus, 9(1): 18-27, 1957.

Röhl, U., Bralower, T. J., Norris, R. D., and Wefer, G.: New chronology for the late Paleocene thermal maximum and its environmental implications, Geology, 28, 927-930, 10.1130/0091-7613(2000)28<927:ncftlp>2.0.co;2, 2000.

Röhl, U., Westerhold, T., Bralower, T. J., and Zachos, J. C.: On the duration of the Paleocene-Eocene thermal maximum (PETM), Geochemistry, Geophysics, Geosystems, 8, 10.1029/2007GC001784, 2007.

Schankler, D.: Faunal zonation of the Willwood Formation in the central Bighorn Basin, Wyoming., in: Early Cenozoic paleontology and stratigraphy of the Bighorn Basin, Wyoming, edited by: Gingerich, P. D., University of Michigan Papers on Paleontology, Ann Arbor, MI, 99-114, 1980.

Sluijs, A., Bowen, G. J., Brinkhuis, H., Lourens, L. J., and Thomas, E.: The Palaeocene–Eocene Thermal Maximumsuper greenhouse: biotic and geochemical signatures, age models and mechanisms of global change, in: Deep-Time Perspectives on Climate Change: Marrying the Signal from Computer Models and Biological Proxies, edited by: Williams, M., Haywood, A. M., Gregory, F. J., and Schmidt, D. N., The Micropalaeontological Society Special Publications, The Geological Society, London, 323-349, 2007.

Stap, L., Sluijs, A., Thomas, E., and Lourens, L.: Patterns and magnitude of deep sea carbonate dissolution during Eocene Thermal Maximum 2 and H2, Walvis Ridge, southeastern Atlantic Ocean, Paleoceanography, 24, 10.1029/2008PA001655, 2009.

Thomas, E.: Development of Cenozoic deep-sea benthic foraminiferal faunas in Antarctic waters, in: Origins and Evolution of the Antarctic Biota, edited by: Crame, J. A., Geological Society Special Publication, 283-296, 1989.

Thomas, E., and Shackleton, N. J.: The Paleocene-Eocene benthic foraminiferal extinctio and stable isotope anomalies, in: Correlationof the Early Paleogene in Northwest Europe, edited by: Knox, R. W. O. B., Corfield, R. M., and Dunay, R. E., Geological Society Special Publication, 401-441, 1996.

Westerhold, T., Röhl, U., Laskar, J., Bowles, J., Raffi, I., Lourens, L. J., and Zachos, J. C.: On the duration of magnetochrons C24r and C25n and the timing of early Eocene global warming events: Implications


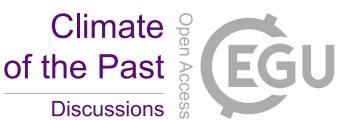

from the Ocean Drilling Program Leg 208 Walvis Ridge depth transect, Paleoceanography, 22, 10.1029/2006PA001322, 2007.

Westerhold, T., Röhl, U., Raffi, I., Fornaciari, E., Monechi, S., Reale, V., Bowles, J., and Evans, H. F.: Astronomical calibration of the Paleocene time, Palaeogeography, Palaeoclimatology, Palaeoecology,
257, 377-403, 10.1016/j.palaeo.2007.09.016, 2008.

Westerhold, T., Röhl, U., Frederichs, T., Bohaty, S. M., and Zachos, J. C.: Astronomical calibration of the geological timescale: closing the middle Eocene gap, Clim. Past Discuss., 11, 1665-1699, 10.5194/cpd-11-1665-2015, 2015.

Westerhold, T., Röhl, U., Frederichs, T., Agnini, C., Raffi, I., Zachos, J. C., and Wilkens, R. H.:
Astronomical Calibration of the Ypresian Time Scale: Implications for Seafloor Spreading Rates and the Chaotic Behaviour of the Solar System?, Clim. Past Discuss., 2017, 1-34, 10.5194/cp-2017-15, 2017.

Wilkens, R., Westerhold, T., Drury, A. J., Lyle, M., Gorgas, T., and Tian, J.: Revisiting the Ceara Rise, equatorial Atlantic Ocean: isotope stratigraphy of ODP Leg 154, Climate of the Past Discussions,
2017, 1-22, doi: 10.5194/cp-2016-140, 2017.

Zachos, J. C., Röhl, U., Schellenberg, S. A., Sluijs, A., Hodell, D. A., Kelly, D. C., Thomas, E., Nicolo, M., Raffi, I., Lourens, L. J., McCarren, H., and Kroon, D.: Rapid Acidification of the Ocean During the Paleocene-Eocene Thermal Maximum, Science, 308, 1611-1615, 10.1126/science.1109004, 2005.

Zachos, J. C., McCarren, H., Murphy, B., Röhl, U., and Westerhold, T.: Tempo and scale of late Paleocene
and early Eocene carbon isotope cycles: Implications for the origin of hyperthermals, Earth and Planetary Science Letters, 299, 242-249, 10.1016/j.epsl.2010.09.004, 2010.

Zeebe, R. E., and Zachos, J. C.: Reversed deep-sea carbonate ion basin gradient during Paleocene-Eocene thermal maximum, Paleoceanography, 22, 10.1029/2006PA001395, 2007.

Zeebe, R. E., Ridgwell, A., and Zachos, J. C.: Anthropogenic carbon release rate unprecedented during
the past 66 million years, Nature Geosci, 9, 325-329, 10.1038/ngeo2681, 2016.



**Table 1**. **PETM Age Model for Polecat Bench (PCB) and ODP Sites 690, 1262, 1263, 1265, 1266, and 1267.** From left to right: Precession cycle number of the extracted cycles as in Röhl et al. 2007 and this study, the equivalent depth in the Polecat Bench outcrop and the BBCP drill core, the respective depth of the assigned precession cycles in ODP Sites, the relative age to the onset of the PETM assuming 21 kyr duration for each precession cycle counted, and finally the absolute age using 55.930 Ma as the age of the onset of the PETM as in Westerhold et al. 2007.

| Precession cycle No. | | PCB outcrop depth AA08 | PCB mcd this study | Depth | | | | | | Age rel. to onset PETM (Pre=21.0 kyr) | Age PETM @ 55.930 Ma |
| Röhl et al., 2007 | this study | | | Site 1262 mcd | Site 1263 rmcd | Site 1265 mcd | Site 1266 mcd | Site 1267 rmcd | Site 690B mbsf | | |
|---|---|---|---|---|---|---|---|---|---|---|---|
| 20 | 20 | | | 135.88 | 326.56 | 308.24 | | 227.12 | 160.42 | 409 | 55.521 |
| 19 | 19 | | | 136.09 | 327.01 | 308.66 | | 227.35 | 160.98 | 388 | 55.542 |
| 18 | 18 | | | 136.31 | 327.53 | 309.19 | | 227.62 | 161.57 | 367 | 55.563 |
| 16+17 | 17 | | | 136.70 | 328.30 | 309.85 | | 228.00 | 162.40 | 346 | 55.584 |
| 15 | 16 | | | 137.04 | 329.11 | 310.44 | | 228.37 | 163.54 | 325 | 55.605 |
| 14 | 15 | | 7.90 | 137.27 | 329.58 | 310.81 | 302.05 | 228.70 | 164.32 | 304 | 55.626 |
| 13 | 14 | | 16.10 | 137.50 | 330.13 | 311.29 | 302.50 | 228.95 | 164.87 | 283 | 55.647 |
| 12 | 13 | | 23.50 | 137.75 | 330.78 | 311.76 | 302.95 | 229.25 | 165.29 | 262 | 55.668 |
| 11 | 12 | | 31.20 | 138.12 | no data | 312.31 | 303.52 | 229.57 | | 241 | 55.689 |
| 10 | 11 | | 38.20 | 138.38 | no data | 313.01 | 304.10 | 229.90 | 166.24 | 220 | 55.710 |
| 9 | 10 | | 44.60 | 138.73 | 332.12 | 313.64 | 304.70 | 230.30 | | 199 | 55.731 |
| 8 | 9 | | 51.00 | 139.03 | 332.79 | 314.21 | 305.12 | 230.64 | 167.20 | 178 | 55.752 |
| 7 | 8 | 56.95 | 58.20 | 139.32 | 333.39 | 314.66 | 305.52 | 230.93 | | 157 | 55.773 |
| 6 | 7 | 67.55 | 65.85 | 139.53 | 333.88 | 315.23 | 305.93 | 231.22 | | 136 | 55.794 |
| 5 | 6 | 75.45 | 74.40 | 139.71 | 334.29 | 315.45 | 306.21 | 231.40 | 169.29 | 115 | 55.815 |
| 4 | 5 | 83.35 | 82.60 | 139.82 | 334.57 | 315.59 | 306.40 | 231.52 | 169.65 | 94 | 55.836 |
| 3 | 4 | 91.90 | 91.10 | 139.94 | 334.82 | 315.72 | 306.56 | 231.63 | 169.98 | 73 | 55.857 |
| 2 | 3 | 99.95 | 99.70 | 140.03 | 335.04 | 315.80 | 306.67 | 231.69 | 170.30 | 52 | 55.878 |
| 1 | 2 | 107.85 | 107.50 | | | | | | | 31 | 55.899 |
| missing | 1 | 114.85 | 115.35 | | | | | | | 10 | 55.920 |
| **onset PETM** | | | **118.70** | **140.14** | **335.27** | **315.85** | **306.77** | **231.77** | **170.64** | **0** | **55.930** |
| -1 | -1 | 126.75 | 123.00 | 140.23 | 335.64 | 316.31 | 306.92 | | 170.99 | -10 | 55.940 |
| -2 | -2 | | 130.00 | 140.32 | | 316.74 | 307.20 | 232.05 | 171.40 | -31 | 55.961 |
| -3 | -3 | | 139.30 | 140.53 | | | | 232.27 | 171.88 | -52 | 55.982 |
| -4 | -4 | | 146.50 | | | | | 232.45 | 172.40 | -73 | 56.003 |
| -5 | -5 | | 155.00 | 140.85 | 336.86 | | 308.10 | 232.63 | 173.14 | -94 | 56.024 |
| -6 | -6 | | 164.00 | 141.04 | 337.11 | | 308.43 | 232.85 | 173.54 | -115 | 56.045 |
| -7 | -7 | | 171.20 | 141.27 | 337.42 | | 308.80 | 233.10 | 173.73 | -136 | 56.066 |
| -8 | -8 | | 181.70 | 141.52 | 337.82 | | 309.13 | 233.35 | 174.11 | -157 | 56.087 |
| -9 | -9 | | 189.50 | 141.74 | 338.20 | | 309.55 | 233.65 | | -178 | 56.108 |
| -10 | -10 | | | 141.95 | 338.47 | | 309.87 | 233.88 | | -199 | 56.129 |
| -11 | -11 | | | 142.12 | | | | | | -220 | 56.150 |
| -12 | -12 | | 208.70 | 142.41 | | | | | | -241 | 56.171 |
| -13 | -13 | | | 142.63 | | | | | | -262 | 56.192 |
| -14 | -14 | | | 142.87 | | | | | 174.34 | -283 | 56.213 |
| -15 | -15 | | 228.60 | 143.10 | | | | | 174.58 | -304 | 56.234 |
| -16 | -16 | | 235.70 | 143.39 | | | | | 174.99 | -325 | 56.255 |
| -17 | -17 | | 243.50 | 143.71 | | | | | | -346 | 56.276 |
| -18 | -18 | | 251.60 | 143.96 | | | | | 175.65 | -367 | 56.297 |
| -19 | -19 | | | 144.27 | | | | | 176.05 | -388 | 56.318 |
| -20 | -20 | | | 144.60 | | | | | 176.45 | -409 | 56.339 |



**Table 2. Age Model for Gilmore Hill (GMH) drill core and regional outcrops.** From left to right: Labeling ID for identified precession cycles (see text for details), the respective depth of the assigned precession cycles in the Upper Deer Creek, the Deer Creek and the Gilmore Hill sections as well as the Gilmore Hill BBCP drill core. Precession cycle number as defined in ODP sites counting precession cycles (Westerhold et al. 2007) followed by the relative and absolute age to the onset of the PETM assuming 21 kyr duration for each precession cycle counted, using 55.930 Ma for the onset of the PETM (Westerhold et al. 2007) and using 54.050 Ma for the absolute age of ETM-2 (Westerhold et al. 2017).

| ID | Upper Deer Creek meter Abels et al. 2012 | Deer Creek meter Abels et al. 2013 | Gilmore Hill meter DAmbrosia et al. 2017 | GMH mcd | precession cycle no | relative to | | |
|---|---|---|---|---|---|---|---|---|
| | | | | | | onset PETM @ 0 | Onset PETM @ 55.93 Ma | ETM-2 @54.05 Ma |
| H2 | 98.05 | - | | - | - | - | - | 53.950 |
| *End of Elmo CIE* | | | 920.0 | | 90 | 1788 | 54.058 | 54.008 |
| Elmo | 70.00 | - | 907.0 | 10.00 | 88 | 1830 | 54.100 | 54.050 |
| Y | 42.29 | - | - | 26.55 | 87 | 1809 | 54.121 | 54.071 |
| X | 36.30 | - | - | 33.90 | 86 | 1788 | 54.142 | 54.092 |
| W | 29.63 | - | 854.5 | 41.10 | 85 | 1767 | 54.163 | 54.113 |
| V | 23.81 | - | - | 48.00 | 84 | 1746 | 54.184 | 54.134 |
| U | - | - | - | 54.40 | 83 | 1725 | 54.205 | 54.155 |
| T | - | - | - | 60.40 | 82 | 1704 | 54.226 | 54.176 |
| S | - | - | 824.0 | 66.80 | 81 | 1683 | 54.247 | 54.197 |
| R | - | - | - | 71.40 | 80 | 1662 | 54.268 | 54.218 |
| Q | - | - | - | 77.20 | 79 | 1641 | 54.289 | 54.239 |
| P | - | 110.9 | - | 84.80 | 78 | 1620 | 54.310 | 54.260 |
| O | - | 103.8 | 796.5 | 92.70 | 77 | 1599 | 54.331 | 54.281 |
| N | - | 96.8 | - | 98.40 | 76 | 1578 | 54.352 | 54.302 |
| M | - | - | - | 105.60 | 75 | 1557 | 54.373 | 54.323 |
| L | - | 79.5 | - | 112.70 | 74 | 1536 | 54.394 | 54.344 |
| K | - | 72.6 | - | 120.70 | 73 | 1515 | 54.415 | 54.365 |
| J | - | 65.8 | 769.0 | 127.80 | 72 | 1494 | 54.436 | 54.386 |
| I | - | 59.0 | - | 134.80 | 71 | 1473 | 54.457 | 54.407 |
| H | - | 52.3 | - | 141.60 | 70 | 1452 | 54.478 | 54.428 |
| G | - | 46.0 | - | 153.60 | 69 | 1431 | 54.499 | 54.449 |
| F | - | - | - | - | 68 | 1410 | 54.520 | 54.470 |
| E | - | 28.2 | - | 174.00 | 67 | 1389 | 54.541 | 54.491 |
| D | - | 21.3 | - | 179.70 | 66 | 1368 | 54.562 | 54.512 |
| C | - | 15.0 | - | 187.30 | 65 | 1347 | 54.583 | 54.533 |
| B | - | 9.3 | - | 194.70 | 64 | 1326 | 54.604 | 54.554 |
| A | - | 3.2 | - | 200.30 | 63 | 1305 | 54.625 | 54.575 |
| -A | - | - | - | 205.90 | 62 | 1284 | 54.646 | 54.596 |



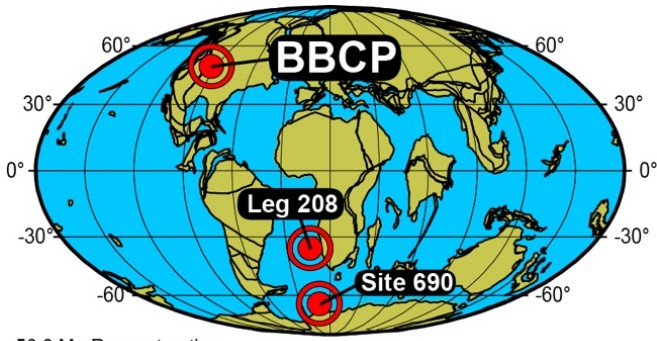

**Figure 1: Location map for ODP sites 702, 1260 and 1263 on a 40 Ma paleogeographic reconstruction in Mollweide projection (from http://www.odsn.de).**



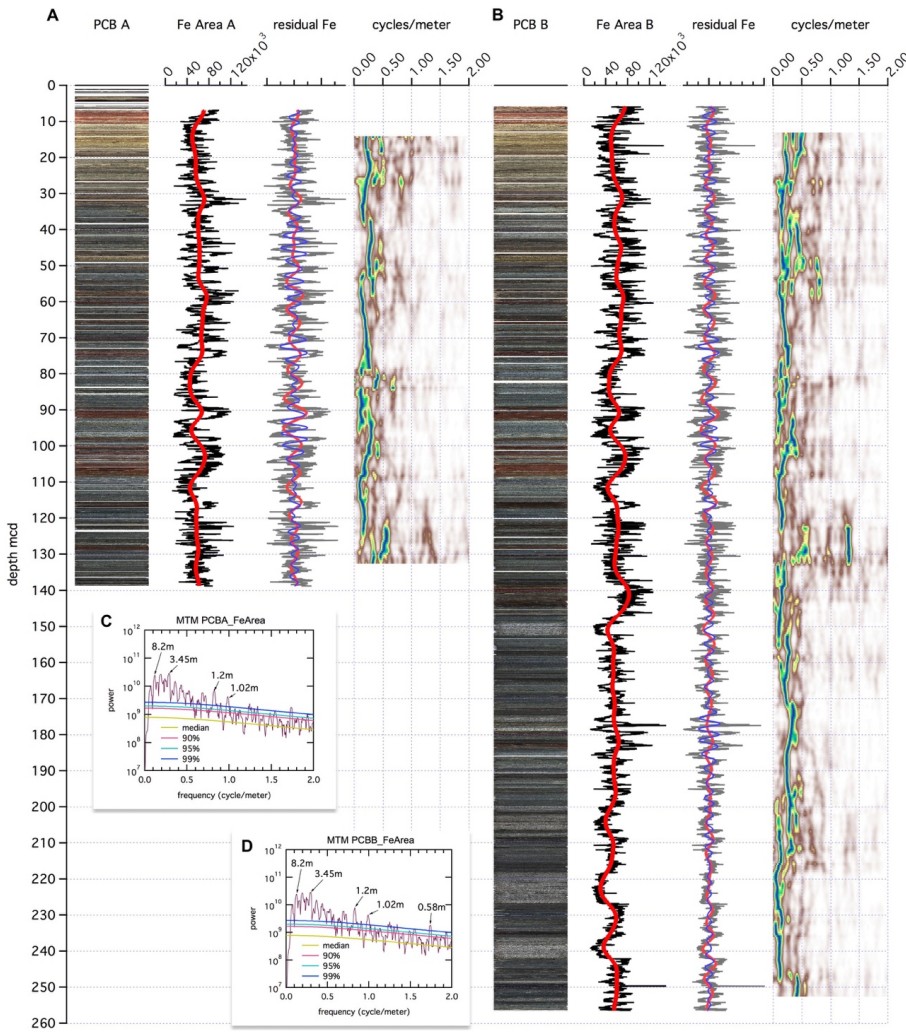

**Figure 2: Polecat Bench PCB-A (A) and PCB-B (B) core images, XRF Fe intensity data and spectral analysis on composite depth scale. Core scan images have been assembled with software package IGOR (Wilkens et al. subm). XRF Fe intensity (black line) with trend (thick red line) that has been removed for following spectral analysis. Residual Fe after trend removal and two Gaussian filters of the dominant cycles 8m (red) and 3.5m (blue). Evolutive spectral plot to decode changes in the cyclicity and thus sedimentation rates. Panels (C) and (D) show the MTM-power spectra for PCB-A and PCB-B Fe data.**



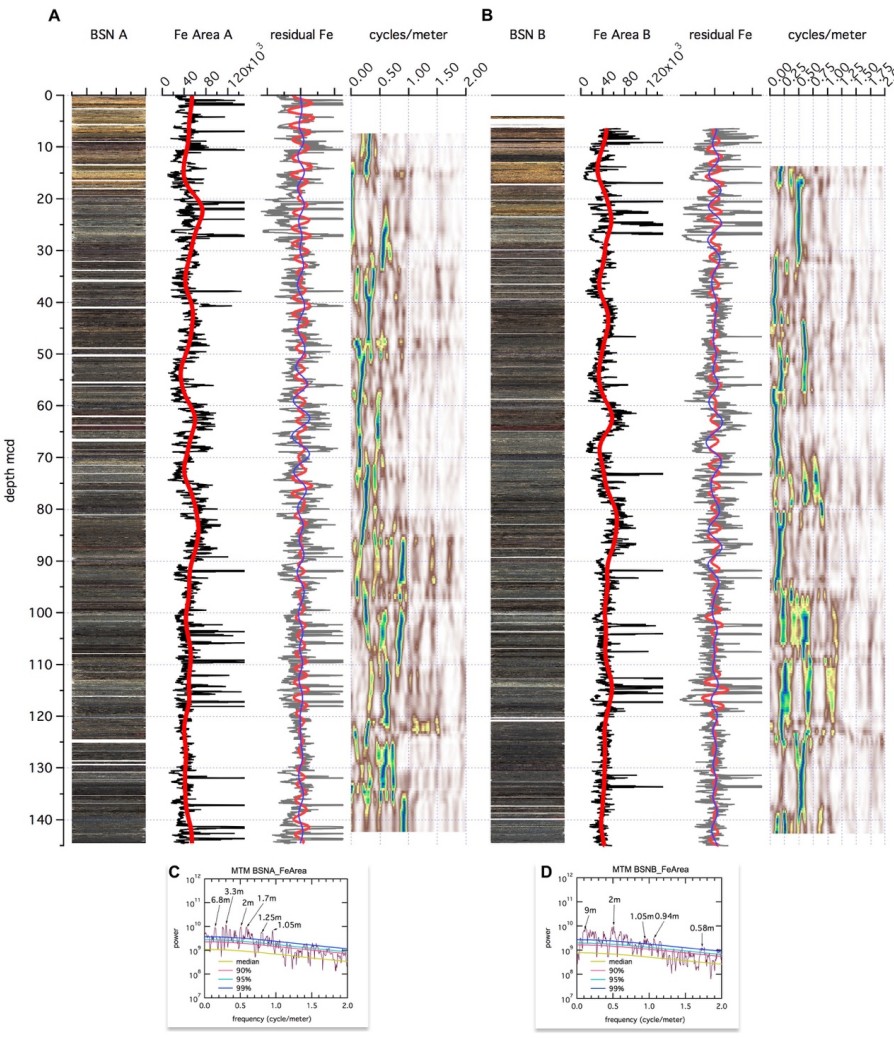

**Figure 3: Basin Substation BSN-A (A) and BSN-B (B) core images, XRF Fe intensity data and spectral analysis on composite depth scale. Panels (C) and (D) show the MTM-power spectra for BSN-A and BSN-B Fe data. For details see figure 2.**



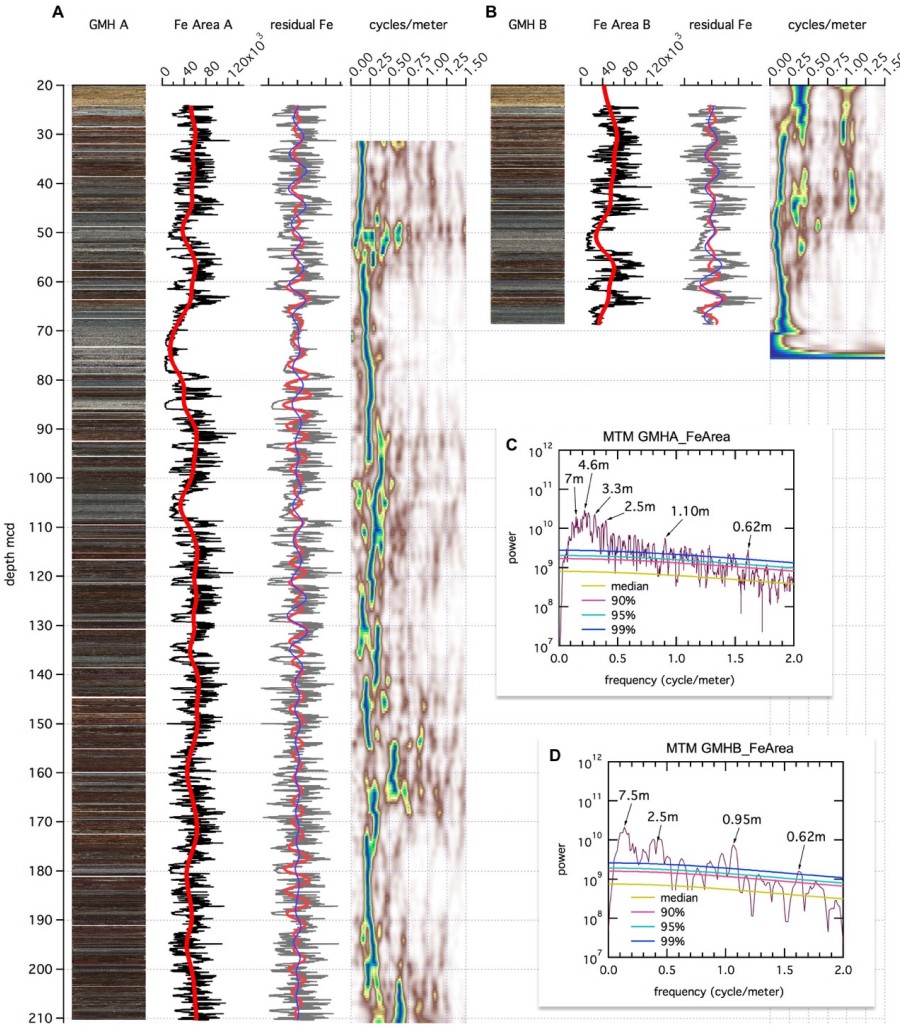

**Figure 4:** Gilmore Hill GMH-A (A) and GMH-B (B) core images, XRF Fe intensity data and spectral analysis on composite depth scale. Panels (C) and (D) show the MTM-power spectra for GMH-A and GMH-B Fe data. For details see figure 2.





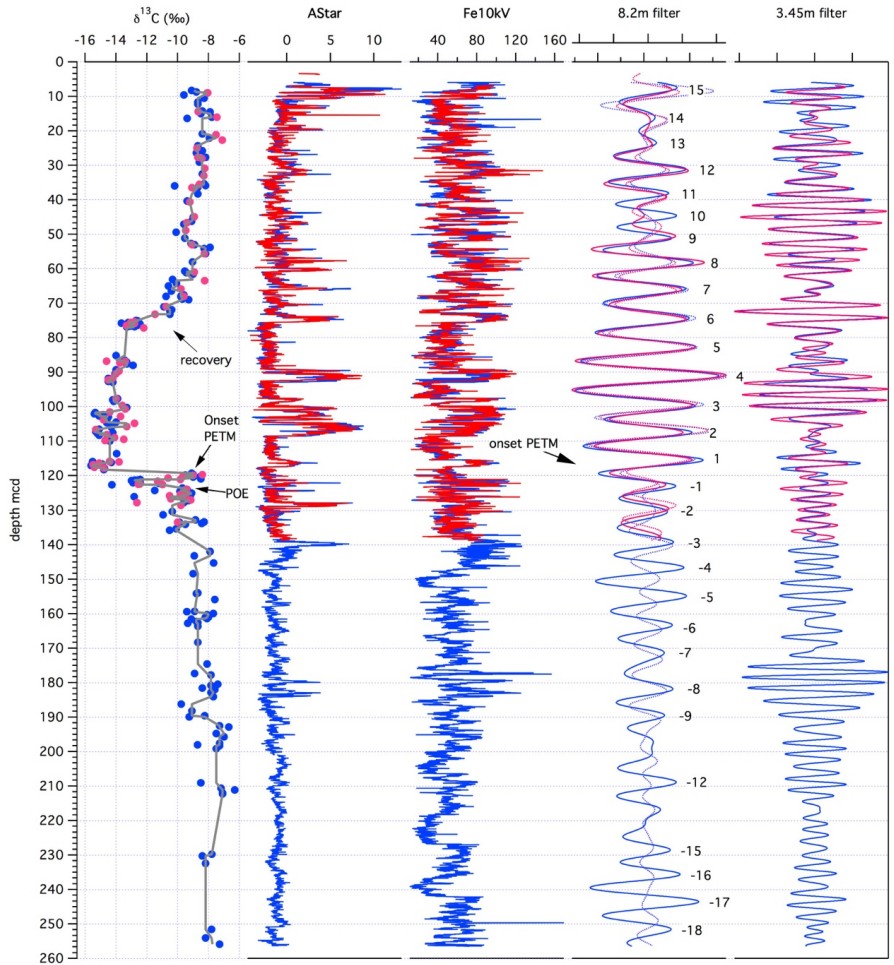

**Figure 5: Cyclostratigraphy for Polecat Bench. From left to right: PCB-A(red) and PCB-B (blue) soil nodule carbon isotope data (Bowen et al. 2015), a\* data from color scanning, XRF Fe intensity data (in total counts area \*1000). Then Gaussian filter of the longer 8.2 m cycle (precession) of the Fe data (lines) and from a\* data (dashed lines). Numbers mark the precession cycle counting starting at the PETM, positive numbers is time after PETM, negative numbers before. On the right, Gaussian filter of the 3.5 m cycle (half-precession).**



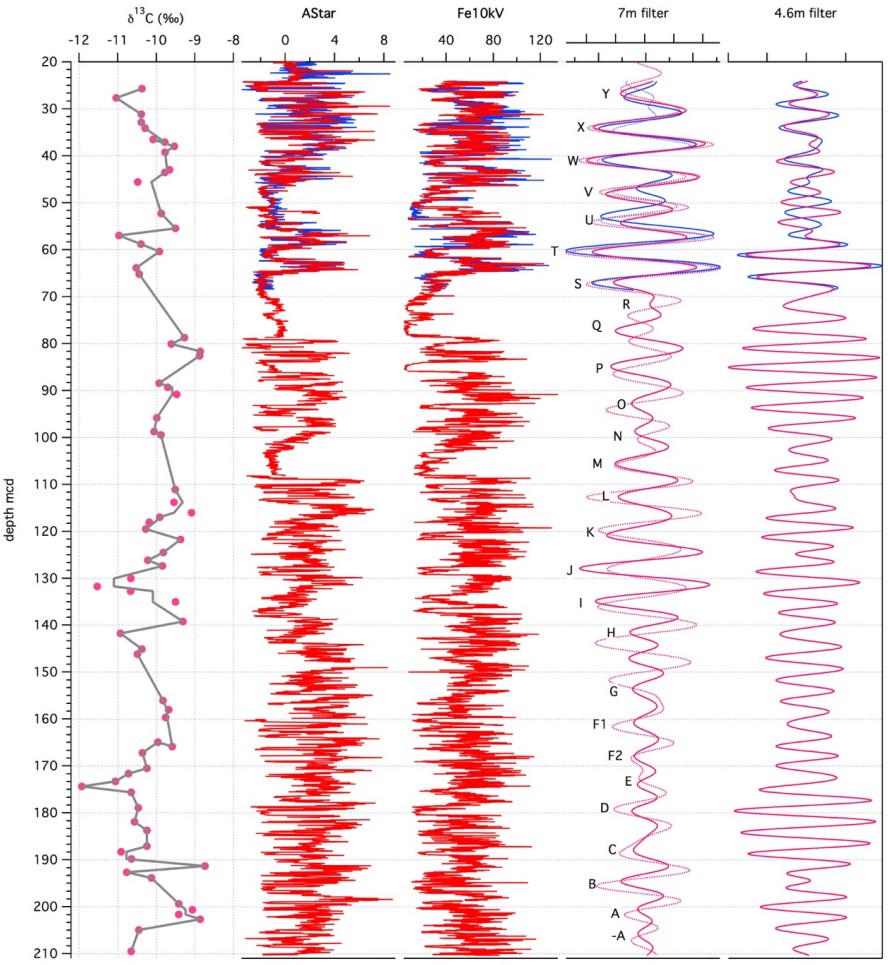

**Figure 6: Cyclostratigraphy for Gilmore Hill. Left to right: GMH A (red) soil nodule carbon isotope data (Tab. S9) for stratigraphic reference, a\* data from color scanning, XRF Fe intensity data (in total counts area \*1000). Then Gaussian filter of the longer 7 m cycle (precession) of the Fe data (lines) and from a\* data (dashed lines). Letters mark the precession cycle counting following the Abels et al. (2013) labeling at Deer Creek Amphitheater. On the right, Gaussian filter of the 4.6 m cycle (half-precession). Please note: the cyclostratigraphy is straight forward and correlates well to Abels et al. (2013) for the range A to P, however, cycles P to Y are new and not in Abels et al. (2013).**





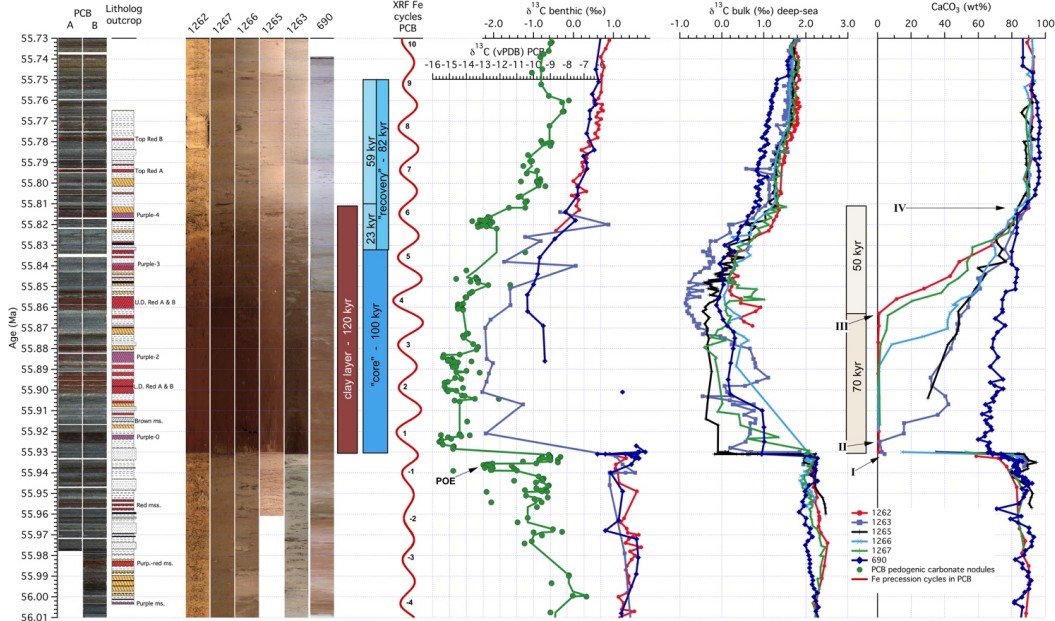

**Figure 7: Overview for the Paleocene-Eocene Thermal Maximum (PETM) data from deep-sea records and the terrestrial Polecat Bench (PCB) drill core against age.** Core images and lithology log (Gingerich et al., 2006) for PCB, core images of ODP Sites 1262, 1267, 1266, 1265, 1263 and 690 (aligned from left to right according to the water depth from deep to shallow), defined phases of events in the PETM (Röhl et al., 2007) on the new age model, extracted Gaussian filter of the PCB XRF Fe intensity data, stable carbon isotope data from PCB soil nodules (Bowen et al., 2015) and the deep sea benthic foraminifera and bulk sediment (690 -Bains et al., 1999; Leg 208 - Zachos et al., 2005), and carbonate content (690 - Farley and Eltgroth, 2003; Leg 208 - Zachos et al., 2005). Letters indicate horizons as identified by Zachos et al., (2005) adjusted to the new age model for the deep-sea sites.



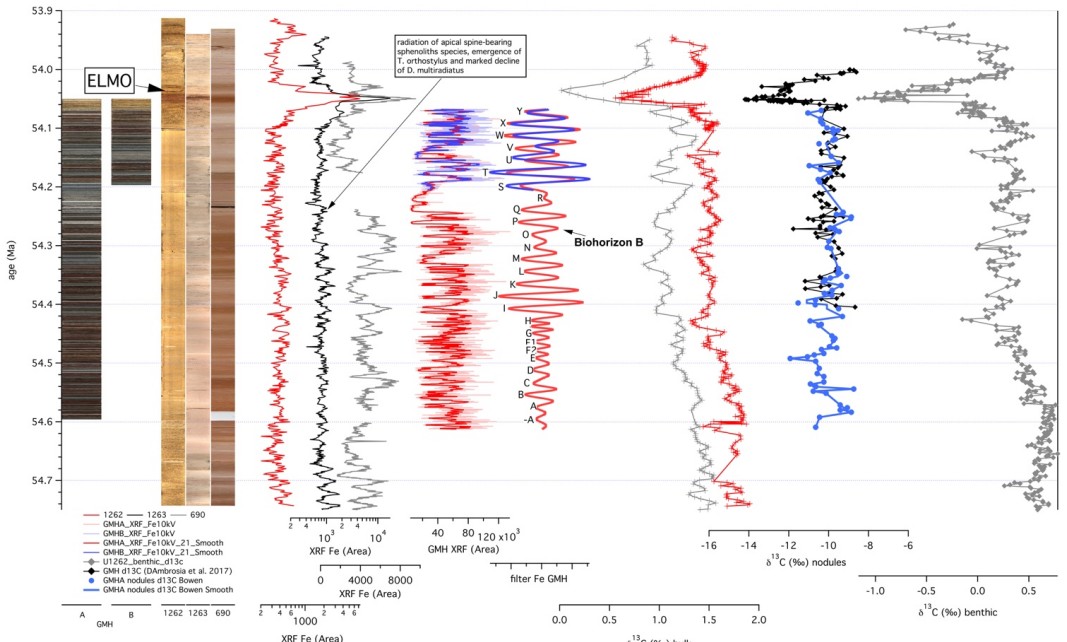

**Figure 8: Overview for the interval prior to the Eocene Thermal Maximum 2 (ETM-2) data from deep-sea records and the terrestrial Gilmore Hill (GMH) drill core against age. Core images for GMH A and B, core images of ODP Sites 1262, 1263 and 690 (aligned from left to right according to the water depth from deep to shallow), XRF Fe core scanning data from 1262 (red), 1263 (black), 690 (grey) (Westerhold et al., 2007), and GMH, extracted Gaussian filter of the GMH XRF Fe intensity data, stable carbon isotope data of soil nodules from the Gilmore Hill area (black – Gilmore Hill section Abels et al., 2012 and D'Ambrosia et al., 2017; blue – GMH drill core) and the deep sea benthic foraminifera (1262 – Littler et al., 2014) and bulk sediment (690 – Cramer et al., 2003; 1262 - Zachos et al., 2010). Position of Biohorizon B is after Schlanker (1980) and Chew (2009), the change in calcareous nannofossils (box) in ODP Site 1262 from Agnini et al., (2007).**