# Peer review of "Synchronizing early Eocene deep-sea and continental records – cyclostratigraphic age models for the Bighorn Basin Coring Project drill cores"

_Climate of the Past, 2017_

## Author Comment (AC1) · 15 Jun 2017

Dear Reviewer,

All data generated within this study will be open access at the PANGAEA database if the manuscript is accepted for publication.

For the review process in the discussion phase we sent the entire dataset to the editor. Reviewers can request the dataset if needed from the editor upon request.

Kind Regards, Thomas Westerhold

---

## Referee Comment (RC1) · Anonymous Referee #1 · 5 Dec 2017

The aim of the ms is to construct astrochronologic age models for the Bighorn Basin Coring Project and to compare and synchronize these age models with those of the deep-sea. This is a valid approach as the continental succession of the Bighorn Basin is marked by high sedimentation rates and the number of precession related cycles might be more easy to count than in the deep-sea records with their associated CaCO3 dissolution especially during the PETM. The "new" age models are not fundamentally different from published age models based on outcrop. Yet, it is timely to compare them in detail with the marine models, also in view of discrepancies with independent He3 based age models. The ms. can probably be accepted after major revision but certain issues definitely need further clarification.

[Figure]

Major points.

Title: Looking at the ms, one may ask why new is included. Are these new cyclostratigraphic age models for the BBCP cores, but then what are the old cyclostratigraphic age models from the project. Moreover, these age models are also not particularly new when compared with the existing age models based on outcrops, as these are largely identical.

MTM spectra 1. The MTM spectra might be somewhat problematical as the null-spectrum and confidence limits do not follow the shape of the spectrum very well. Is this a problem of using the "wrong" model for calculating the null-spectrum (as in Vaughan et al., 2011)? Is it as such logical that the power of all thicknesses between ∼2 and ∼20 m plot above the 99% CL? This band contains the dominant 3.5 and 8 m cycles, but constant power above 99% for such a broad frequency is not very logical.

In this respect, the authors should preferably not use the Mann and Lees (1996; ML96) robust red noise approach that is in SSA-MTM tookit, as it has a tendency to artificially create low frequency cycles (as documented in Meyers, 2012; example in his Figure 2D illustrates that there is a 90% chance of getting false long period cycles from noise). If they want to use the ML96 approach, they should use the one in Astrochron, which fixes this 'edge-effect' problem.

The MTM spectra often show a bewildering numbering of peaks in the frequency band that is of primary interest for the paper. This large number of peaks likely stems from the very long and high-resolution character of the records, but it might be preferable to attempt reducing the number of spectral peaks in this band, as less peaks / resolution imply greater stability of peak position, and is more easy to interpret.

MTM spectra 2. The dramatic reduction of power at very low frequencies in their MTM spectra is due to the data detrending. My concern is that the frequencies they are interested in are up against this detrended region of the spectrum. So if they want to do this detrending, they should also show power spectra without the trends removed,

so one can better evaluate the nature of the peaks (if they are real or a consequence of detrending).

Half-precession. The authors constantly use the term half precession cycle for their 3.5 m cyclicity while their precession related cycles are often more than twice as thick. This problem was also encountered by Abdul-Aziz et al. (2008), when studying the Polecat Bench and Red Butte sections; they concluded that this cycle, which is related to prominent individual paleosols, does not represent half (or semi-) precession, but has a period that is significantly shorter and closer to that of the Heinrich events of the last 100.000 year. This was confirmed by the results of bandpass filtering, and the same is the true for the results of the filtering in the present ms (see their Figure 5 where more than two cycles fit into one precession-related cycle). Hence, the authors should use sub-precession or millennial-scale rather than half-precession.

Precession minimum (l. 24, p. 9). The authors mention that the PETM onset was in a precession minimum, but it is not clear where that comes from (they refer to see above). Do they mean the Fe- minimum between their cycle 1 and -1? But where is the phase relation with precession based on, or it this a mistake? It is also not perfectly clear to me how that might explain the 4-kyr discrepancy in addition of the one precession cycle misfit, also because discrepancies of 4-kyr are within the uncertainty of all the age models.

Comparison between cyclostratigraphic and Helium-based age models. (l.10-28, p.10). This discussion is becoming a bit semantic and potentially far-fetched, going into much detail, which may not all be that relevant to explain the observed major offset of 40-kyr for the initial rapid recovery of the CIE. Also, the average duration of a precession cycle remains an average and longer precession periods are generally found in intervals with high eccentricity (maxima) and shorter cycles during eccentricity minima (as the different cycles are differently modulated), so this plays a role as well. However, again the difference might be either too small to explain the offset or will only (slightly) enhance it. A relevant question to ask is whether the Helium isotope ratio is not affected (or not)

by the enhanced volcanic activity at that time as the East Greenland flood basalts may have formed at the same time (see Wotzlaw et al.).

Almost. A very interesting and also intriguing aspect is the potential causal connection between Biohorizon B in the Bighorn Basin and the calcareous nannofossil events of the same age in the marine realm, intriguing especially as the proxies in the marine record do not indicate that something dramatic is happening. However, the authors use the curious wording almost when comparing the continental faunal turnover with the marine events. But almost is not of the same age, so what is exactly the difference and how does almost the same age translate to potentially having the same origin. In Figure 8, the age difference between the two events is ∼40-kyr, but what are the uncertainties in the respective age models and in the position and thus age of the respective bio-events? The uncertainty in the position of Biohorizon B might be quite large compared with that in the marine record, so do these uncertainties overlap? The reason to develop high-resolution astrochronologic age models is meant to increase the temporal resolution and solve possible temporal relationships and chicken-and-egg problems. So what does this almost imply in this case? This should be made more clear in the ms.

Minor points.

Cycle 0. Why is there no cycle 0 in the numbering of the cycles in the PCB cores

405-kyr minimum (p.7, l. 14). Is this also not part of a very long 2.0 Myr eccentricity cycle, see Lourens et al., (2005) and Meyers (2015)? Please check what you mean exactly.

3.2 Time series analysis of BBCP drill cores. The first paragraph before 3.2.1 belongs to the Material & Methods section rather than to Results.

---

## Referee Comment (RC2) · Anonymous Referee #2 · 6 Dec 2017

Review of

**"Synchronizing early Eocene deep-sea and continental records – new cyclostratigraphic age models from the Bighorn Basin Coring Project"**

by **Westerhold and co-authors**
submitted to **'Climate of the Past'** (Initial submission, august 2017)
Manuscript No: **cp-2017-74**

**Overview:**

Westerhold and co-authors performed a 'late Paleocene to early Eocene' cyclostratigraphic study on sedimentary terrestrial (fluvial-paleosols) records from continental drill cores of the Bighorn Basin Coring Project (BBCP, Wyoming, USA), then they compared the inferred cyclostratigraphic timescale to previous timescales obtained from sedimentary marine (deep-sea) records, from ocean drilling programs (ODP).

The main purpose of Westerhold et al.'s study is to face cyclostratigraphic timescales from continental and marine realms in order to reconcile the duration of the PETM. Secondarily, Westerhold and co-authors proposed an astronomical timescale for the interval prior to the Elmo (ETM2) event, then discussed a potential link between "Biohorizon B" mammalian turnover and the perturbation in the marine biotic cycle (in calcareous nannofossil assemblages, in particular).

For cyclostratigraphy, Westerhold and co-authors used high-resolution XRF iron (Fe) scanning data and sediment color redness (a* reflectance) together with previously published soil nodule $\delta^{13}C$ data in order to estimate the duration of the PETM (from Polecat Bench and Basin Substation cores), and secondarily to establish an orbital timescale for the interval prior to the Elmo event (from Gilmore Hill cores).

A highly resolved 'late Paleocene to early Eocene' cyclostratigraphic timescale is needed to enhance our comprehension of the associated abrupt, severe changes in Earth's surface systems at the PETM and Elmo events.

Overall, the manuscript is well written and structured, and the discussion with previous age models was well conducted, though some improvements should be done (see 'specific comments'). All of these qualities make the manuscript a significant contribution to '*Climate of the Past*'.

Below are some specific (but not major) points that I classified into five items, and minor points taken out from the text. Among the specific points I raised, two capital issues should be clearly addressed, which are related to: (i) the additional precession cycle to the clay-layer interval, and (ii) the definition of the end of CIE in both deep-sea and terrestrial realms.

**Specific comments:**

**1) The definition of the end of CIE-PETM in the Polecat Bench $\delta^{13}C$ record.**

Duration estimates of the PETM and comparison with previous studies depend tightly on how the stratigraphic extent of CIE is defined. While it is easier to define the onset of CIE at both realms, the definition of its end is more problematic, especially in the continental $\delta^{13}C$ record.

In the present form of the ms, it is not clear how the authors (or may be by referrig to previous papers) set the end of CIE in the $\delta^{13}C$ terrestrial record. Based on their figure 5, I can place it appropriately at 60 mcd depth, and largely at 55 mcd depth. This implies respectively 7.5 to 8 precession cycles, yielding respective durations of 157.5 and 168 kyr (21 kyr mean precession period). These durations are close to the 171 kyr estimate inferred from deep-sea records (Röhl et al., 2007).

In all figures dealing with CIE's correlation between terrestrial and deep-sea records (Figures 7, S12 and S13), the onset of CIE is clearly shown at the abrupt negative $\delta^{13}C$ shift, whereas the end of CIE is not obvious neither at ODP sites nor in PCB terrestrial record. It's even sometimes confused

when reading the cyclostratigraphic interpretation against the proposed age model, and what is said in the text. For instance, in figure 7, the duration of the entire CIE is assessed at about 180 kyr (120 kyr for the clay layer indicated by the brown rectangle plus 59 kyr till the end of CIE shown by light blue rectangle). In the text, the authors discuss a longer duration of 200 kyr..

Again, considering a very likely end of CIE in the terrestrial $\delta^{13}$C data at the top (maximum) of precession cycle no. 8 (Fig. 7), a duration of 168 ky (21 kyr x 8 cycles) could be inferred...

A focus was also given on the duration of clay-layer interval. The clay layer is characteristic of deep-sea environment. What is the degree of reliability of correlation between terrestrial and deep-sea (using $\delta^{13}$C) data that led to the projection of equivalent clay-layer interval into the terrestrial records? Note that this correlation is crucial for the assessment of duration of the clay layer. Could the authors add uncertainties on their stratigraphic correlation?

In summary, the authors should state clearly in the manuscript how they define the stratigraphic extent of the entire CIE (especially its end) and the projected clay-layer into the terrestrial records, and accordingly they could compare duration estimates between the two realms.

**2) Comparison with previous age models**
In the outcrops (Bighorn Basin) in the Polecat Bench section, Abdul-Aziz et al. (2008) arrived to a duration of 157 kyr for the entire CIE-PETM.
Westerhold and co-authors cited Abdul-Aziz et al.'s (2008) study, but they did not explain the 157 kyr shorter duration compared to their longer duration of 200 kyr obtained from Polecat Bench drill cores. Given both studies are based on precession cycle counting from the same basin (and the same Polecat Bench site), I strongly recommend that the authors explicitly discuss the source of such significant difference. Although the authors evoked promptly this difference (Page 7, lines 21-23), but it is still ambiguous how they found a longer duration with regard to a shorter duration provided by Abdul-Aziz et al. (2008) (see also 'Comment 1' above).
Note that Abdul-Aziz et al.'s (2008) duration estimate (i.e., 157 kyr) is close to the 171 kyr duration of Röhl et al. (2007) inferred from deep-sea records.

**3) Amplitude modulation (AM) of the precession by the eccentricity**
The authors outlined 'AM of the precession by the eccentricity' in the text body and they also pointed it out in the abstract and conclusions, however, there is no statistical test (or even an attempt by visual inspection) to show or retrieve such modulation. If the authors would still retain this result, then they should demonstrate it, at least at the short eccentricity band.
The authors stated (Page 7, lines 8 and 9) : "*The filter of the precession cycles of ~8.2 m in both data show modulations that are consistent with eccentricity*". Filtering is not sufficient to draw such conclusion. Here a Hilbert transform is required to extract such AM envelopes...

**4) Half-precession**
Precession vs half-precession ratio is not consistent with the selected bandwidths used for filtering (see for e.g., Fig. 5). Visual inspection in figure 5 indicates that several precession cycles do not match two 'half-precession' cycles, making the hypothesis of 'half-precession' implausible. Also, if the precession central wavelength is 8.2 m, then 'half-precession' central wavelength should be around 4 m (not 3.45 m).
Can the authors resolve this mismatch, by changing the bandwidth for example, or abandon the hypothesis of 'half-precession'.
In addition, the authors stated (Page 5, Lines 28-29) "*The two longer cycles around 8 and 3.5 m have been interpreted as precession and half precession cycles also present in Plio-Pleistocene successions (see Abdul-Aziz et al., 2008)*."
Abdul-Aziz et al. (2008) did not interpret the 3.5 m cycles as half-precession. Instead, they interpreted them as sub-Milankovitch (or millennial). They even stated in their paper « However, the exact origin of sub-Milankovitch cycles remains enigmatic. ». Sub-Milankovitch (or millennial-scale) cycles do not imply half-precession cycles...

**5) Significance of changes in sediment a\* color reflectance and Fe content in terrestrial records**

Although the authors evoked very promptly the potential significance of XRF iron intensity in terrestrial sediments by referring to previous studies (Abels et al., 2012), [and this topic is beyond the scope of the present study], I suggest that the authors develop a little bit the significance of such proxies in terms of climate change (astronomically forced climate). Orbitally driven fluctuations in Fe content in deep-sea sedimentary records have generally (and extensively) been attributed to the relative contribution from carbonate deposition versus detrital-clay inputs. However, the origin of cyclic change in Fe content in terrestrial environments is not well addressed in the litterature...

**Minor points:**

[revised manuscript text omitted]

**Page 3, Line 31:**
"*... the BSN and GMH and PCB sites.*"

Into:

"... the BSN, GMH and PCB sites."

**Page 4, Lines 12 to 26:**
All this paragraph deals with isotope data acquisition, which were already presented in Bowen et al. (2015). Thus, such paragraph should be removed or shortened or moved to the supplementary materials.

**Page 5, Line 15:**
"3.2 Timeseries analysis of BBCP drillcores", please add a hyphen to 'Time-series' and to 'drill-cores' (or a space 'drill core')

**Page 8 (Line 34) and page 9 (Line 1):**
"During the PETM, massive dissolution of carbonates in the deep sea truncated the record (Zachos et al., 2005), complicating the age model constructions."

Into:

"During the PETM, massive dissolution of carbonates in the deep sea truncated the cyclostratigraphic record (Zachos et al., 2005), complicating the construction of age models. "

**Page 9, Lines 31 to 34:**
Adding a precession cycle in deep-sea records to the clay layer is not well argued (see specific comments).
It is likely to miss cycles in XRF Ca records because of the clay layer. However, in Fe and Ba XRF data, cycles are well expressed (see ODP Site 690 in Röhl et al., 2007).

**Page 10, Lines 5 to 7:**
Charles et al. (2011) found 8.5 precession cycles at BH9/05 Core for the entire CIE, similar to Röhl et al. (2007), who used ODP 1263 data.

**Page 11, Line 25:**
"Fe intensities, core images, and color reflectance data were used ...."

into:

"**Sedimentary records** of Fe intensities, core images, and color reflectance data were used ...."

**Page 19 (Figure 1 caption)**

" Location map for ODP sites 702, 1260 and 1263 on a 40 Ma paleogeographic reconstruction in Mollweide projection (from http://www.odsn.de)."

into:

"Location map for BBCP (Bighorn Basin Coring Project, Wyoming, USA), ODP Leg 208 (Sites 1260 and 1263) and ODP Leg 113 (Site 690) on a **56 Ma** paleogeographic reconstruction in Mollweide projection (from http://www.odsn.de, Hay et al., 1999)."

should also refer to the original paper of Hay et al. (1999), and not only to the website

**Page 20 (Figure 2 caption):**
The Wilkens et al.'s (submitted) ms referred in Figure 2 caption, but cited in the reference list as Wilkens et al. (2017); the paper already appeared, so should be 2017.

**Page 23 (Figure 5 caption)**
Line 4: '(lines)' into '(solid lines)'
Line 5: 'at the PETM' into 'at the onset of PETM'

**Page 25 (Figure 7 caption)**
Line 5: "... extracted Gaussian filter of the PCB XRF Fe intensity data"
would better be "... extracted precession cycles using a Gaussian filter of the PCB XRF Fe intensity data".
Could the authors please point the end of the CIE directly on $\delta^{13}C$ data of PCB and deep-sea records?

---

## Author Comment (AC2) · 2 Jan 2018

Dear Referee #1,

Thank you very much for your detailed review of the submitted manuscript.

Below find our reply (red) to your comments (*black*).
* * *
**MAJOR POINTS**

*Title: Looking at the ms, one may ask why new is included. Are these new cyclostratigraphic age models for the BBCP cores, but then what are the old cyclostratigraphic age models from the project. Moreover, these age models are also not particularly new when compared with the existing age models based on outcrops, as these are largely identical.*

As written in the submitted manuscript on page 3, line 11: "The purpose of this report is to establish high-resolution age models for the BBCP drill cores based on cyclostratigraphy and integrate existing age models from outcrops". From this it should be clear that we found on the outcrop "old" age models to develop "new" age models for the drill cores.

To avoid further confusion, we will change the title of the manuscript to "Synchronizing early Eocene deep-sea and continental records – cyclostratigraphic age models for the Bighorn Basin Coring Project drill cores".

*MTM spectra 1. The MTM spectra might be somewhat problematical as the null- spectrum and confidence limits do not follow the shape of the spectrum very well. Is this a problem of using the "wrong" model for calculating the null-spectrum (as in Vaughan et al., 2011)? Is it as such logical that the power of all thicknesses between ~2 and ~20 m plot above the 99% CL? This band contains the dominant 3.5 and 8 m cycles, but constant power above 99% for such a broad frequency is not very logical.*
*In this respect, the authors should preferably not use the Mann and Lees (1996; ML96) robust red noise approach that is in SSA-MTM tookit, as it has a tendency to artificially create low frequency cycles (as documented in Meyers, 2012; example in his Figure 2D illustrates that there is a 90% chance of getting false long period cycles from noise). If they want to use the ML96 approach, they should use the one in Astrochron, which fixes this 'edge-effect' problem.*

All MTM spectra will be replaced by mtmML96 spectra using the Astrochron software package. As an example we show here (Figure 1) the effect mentioned by the referee.

[Figure]

**Figure 1** – Comparison of MTM power spectra for the PCBA core XRF Fe intensities using (a) the SSA-MTM toolkit and (b, c) the Mann and Lees (1996) robust red noise MTM analysis within the *Astrochron* software package. For (b) the raw XRF Fe intensities were used, for (c) the detrended XRF Fe intensities were used as done in (a). Clearly the the ML96 approach is much more appropriate for the data analysis.

*The MTM spectra often show a bewildering numbering of peaks in the frequency band that is of primary interest for the paper. This large number of peaks likely stems from the very long and high-resolution character of the records, but it might be preferable to attempt reducing the number of spectral peaks in this band, as less peaks / resolution imply greater stability of peak position, and is more easy to interpret.*

We agree. Due to the high resolution of the expanded records the spectra show lots of details. Because the spectra have been calculated for the entire length of each record changes in sedimentation rates will produce several closely spaced peaks. To be able to identify changes in sedimentation rates the evolutive spectra were calculated. For the extraction of cycles by filtering of the signal we applied a 30% bandwidth to compensate for these variations in cycle thickness as described in the manuscript.

***MTM spectra 2***. *The dramatic reduction of power at very low frequencies in their MTM spectra is due to the data detrending. My concern is that the frequencies they are interested in are up against this detrended region of the spectrum. So if they want to do this detrending, they should also show power spectra without the trends removed, so one can better evaluate the nature of the peaks (if they are real or a consequence of detrending).*

We will provide MTM spectra using the mtmML96 Astrochron routine for all data. As an example we show the effect of detrending on the data in Figure 1 b and c. The detrending removed cycles longer than 10m.

***Half-precession***. *The authors constantly use the term half precession cycle for their 3.5 m cyclicity while their precession related cycles are often more than twice as thick. This problem was also encountered by Abdul-Aziz et al. (2008), when studying the Polecat Bench and Red Butte sections; they concluded that this cycle, which is related to prominent individual paleosols, does not represent half (or semi-) precession, but has a period that is significantly shorter and closer to that of the Heinrich events of the last 100.000 year. This was confirmed by the results of bandpass filtering, and the same is the true for the results of the filtering in the present ms (see their Figure 5 where more than two cycles fit into one precession-related cycle). Hence, the authors should use sub-precession or millennial-scale rather than half-precession.*

We will correct this in a resubmitted version of the manuscript. However, it is not important for the cyclostratigraphy which is based on the recognition of the precession cycle only.

***Precession minimum*** *(l. 24, p. 9). The authors mention that the PETM onset was in a precession minimum, but it is not clear where that comes from (they refer to see above). Do they mean the Fe- minimum between their cycle 1 and -1? But where is the phase relation with precession based on, or it this a mistake? It is also not perfectly clear to me how that might explain the 4-kyr discrepancy in addition of the one precession cycle misfit, also because discrepancies of 4-kyr are within the uncertainty of all the age models.*

We will clarify this in the revised version of the manuscript. Looking at the initial manuscript figure 5, this can be explained. The onset of the PETM in the PCB core (Bowen et

al. 2015) is between two paleosols showing higher Fe and AStar values at roughly 115 mcd ('purple 0') and 123 mcd. For making an age model we (page 7, line 19): "… also assume that the onset of the PETM is located in the minimum between cycle -1 and 1". Regarding the phase relationship, carbon isotope data from PCB soil nodules (Bowen et al., 2015) in the paleosols are more negative then before and after a paleosol horizon. This is similar to records from the deep-sea around the PETM showing more negative bulk carbonate carbon isotope values in more clay rich layers (higher Fe XRF intensities, Zachos et al. 2010, Littler et al. 2014, Zeebe et al. 2017). Whether the lighter carbon isotopes values and the paleosols correspond to precession minima or maxima (e. g. Lourens et al. 2005) is still unknown, and probably will never be known for certain. This is not relevant for establishing a cyclostratigraphy based on cycle counting.

The onset of the PETM was set into a precession minimum in the PCB cyclostratigraphy model. To correlate deep-sea and terrestrial records, the onset and the top of the initial rapid recovery of the CIE are commonly used (McInerney and Wing, 2011). The PCB cyclostratigraphy indicates that the duration of this interval covers six precession cycles or ~120 kyr (assuming an average duration of 21 kyr for one precession cycle). Therefore, the previous estimate by Röhl et al. 2007 from deep-sea records for this interval of ~95kyr is too short by ~25 kyr or about one precession cycle at the onset of the event. We concluded that a precession cycle is missing in the deep-sea records and needs to be added to those age models. Subtracting 21kyr (one precession cycle) from the missing ~25 kyr leaves us with ~4 kyr. This discrepancy is within any uncertainty of all the age models, as mentioned by the referee. We will clarify this in the revised manuscript.

**Comparison between cyclostratigraphic and Helium-based age models.** *(l.10-28, p.10). This discussion is becoming a bit semantic and potentially far-fetched, going into much detail, which may not all be that relevant to explain the observed major offset of 40-kyr for the initial rapid recovery of the CIE. Also, the average duration of a precession cycle remains an average and longer precession periods are generally found in intervals with high eccentricity (maxima) and shorter cycles during eccentricity minima (as the different cycles are differently modulated), so this plays a role as well. However, again the difference might be either too small to explain the offset or will only (slightly) enhance it. A relevant question to ask is whether the Helium isotope ratio is not affected (or not) by the enhanced volcanic activity at that time as the East Greenland flood basalts may have formed at the same time (see Wotzlaw et al.).*

We thank the referee for this comment. A sentence will be added mentioning the Wotzlaw et al. study. We think discussing or showing the effects of your new age model compared to the Helium age model is important and could be basis for further research, but we refrain from going into too much detail because this is not the scope of our study.

**Almost**. *A very interesting and also intriguing aspect is the potential causal connection between Biohorizon B in the Bighorn Basin and the calcareous nannofossil events of the same age in the marine realm, intriguing especially as the proxies in the marine record do not indicate that something dramatic is happening. However, the authors use the curious wording almost when comparing the continental faunal turnover with the marine events. But almost is not of the same age, so what is exactly the difference and how does almost the*

*same age translate to potentially having the same origin. In Figure 8, the age difference between the two events is ~40-kyr, but what are the uncertainties in the respective age models and in the position and thus age of the respective bio-events? The uncertainty in the position of Biohorizon B might be quite large compared with that in the marine record, so do these uncertainties overlap? The reason to develop high-resolution astrochronologic age models is meant to increase the temporal resolution and solve possible temporal relationships and chicken-and- egg problems. So what does this almost imply in this case? This should be made more clear in the ms.*

This section will be rewritten as the position has been revised slightly including a proper error discussion.

The best constraint on the stratigraphic position of Biohorizon B comes from the Gilmore Hill section where it falls between locality MP167 (LAD of Haplomylus) and MP166 (FAD of Bunophorus), both of which fall directly in the line of section. The mid-level of locality MP167 is at 807 meters (above PETM) of that section and the mid-level of locality MP166 is 840 meters so Biohorizon B must fall somewhere in the interval between ~807 and ~840 meters of the Gilmore Hill section (see Abels et al., 2012 and D'Ambrosia et al., 2017 for details). Another locality, MP122, that is not located directly in the line of section but has been physically correlated to the 825-835 meter level in this section via bed tracing contains both Haplomylus and Bunophorus so provides a more precise biostratigraphic estimate of Biohorizon B but with additional stratigraphic uncertainty due to the long distance correlation. The 33 meters of section between 807 and 840 meters represent 4-5 precession cycles (#79 to #83 in table 2 of the ms) from 54.151 to 54.254 Ma and the 10 meters of section between 825 to 835 meters represent 1.5 precession cycles (between #81 and #83, centered at #82 in table 2 of the ms) from 54.165 To 54.195. We will change the Figure 8 to more accurately represent these uncertainties – see Figure 2 in this reply to the referee.

We will rewrite the chapter and be more critical about temporal uncertainties. However, we think it is worthwhile to point out that the biotic event on land and the deep sea are happening around the same time. It will be made clear that the biotic turnover in the deep-sea is a series of events (representing the fast evolutionary change) rather than a single event. We hope that our manuscript sparks new effort to investigate these biotic turnovers in much more detail to find out how "major" it was.

[Figure]

**Figure 2** – Proposed revision of Figure 8 now showing the interval where the Biohorizon B in the GMH section is located and more details on the series of events occurring in the deep-sea record. The figure is an overview for the interval prior to the Eocene Thermal Maximum 2 (ETM-2) data from deep-sea records and the terrestrial Gilmore Hill (GMH) drill core against age. Core images for GMH A and B, core images of ODP Sites 1262, 1263 and 690 (aligned from left to right according to the water depth from deep to shallow), XRF Fe core scanning data from 1262 (red), 1263 (black), 690 (grey) (Westerhold et al., 2007), and GMH, extracted Gaussian filter of the GMH XRF Fe intensity data, stable carbon isotope data of soil nodules from the Gilmore Hill area (black – Gilmore Hill section Abels et al., 2012 and D'Ambrosia et al., 2017; blue – GMH drill core) and the deep sea benthic foraminifera (1262 – Littler et al., 2014) and bulk sediment (690 – Cramer et al., 2003; 1262 - Zachos et al., 2010). Position of Biohorizon B is after Abels et al., 2012 and D'Ambrosia et al., 2017 (black bar represents best estimate, gray bars represent conservative estimate – see text for discussion); the change in calcareous nannofossils (gray bar and text box) in ODP Site 1262 from Agnini et al., (2007).

**MINOR POINTS**

***Cycle 0****. Why is there no cycle 0 in the numbering of the cycles in the PCB cores*

Why should there be a cycle 0? It is not clear what the referee is pointing at. The onset of the PETM was chosen as the zero line (Table 1 of the ms), in-between precession cycle -1 (before onset PETM) and 1 (after onset PETM).

***405-kyr minimum*** *(p.7, l. 14). Is this also not part of a very long 2.0 Myr eccentricity cycle, see Lourens et al., (2005) and Meyers (2015)? Please check what you mean exactly.*

It is correct as written. The cycles occur in a 405-kyr minimum, a time of low amplitude modulation of the precession cycle by eccentricity.

***3.2 Time series analysis of BBCP drill cores****. The first paragraph before 3.2.1 belongs to the Material & Methods section rather than to Results.*

We would like to keep this section where it is because the time series analysis and thus age model development should ideally be in one chapter.

---

## Author Comment (AC3) · 2 Jan 2018

Dear Referee #2,

Thank you very much for your detailed review of the submitted manuscript.

Below find our reply (red) to your comments (*black*).
* * *
**Specific comments**
***1) The definition of the end of CIE-PETM in the Polecat Bench δ13C record***
*Duration estimates of the PETM and comparison with previous studies depend tightly on how the stratigraphic extent of CIE is defined. While it is easier to define the onset of CIE at both realms, the definition of its end is more problematic, especially in the continental δ13C record.*
*In the present form of the ms, it is not clear how the authors (or may be by referring to previous papers) set the end of CIE in the δ13C terrestrial record. Based on their figure 5, I can place it appropriately at 60 mcd depth, and largely at 55 mcd depth. This implies respectively 7.5 to 8 precession cycles, yielding respective durations of 157.5 and 168 kyr (21 kyr mean precession period). These durations are close to the 171 kyr estimate inferred from deep-sea records (Röhl et al., 2007).*

Definitions for the different phases of the PETM in the deep-sea and the terrestrial realm are given in Zachos et al. 2005, Röhl et al. 2007, Murphy et al. 2010 (all three deep-sea) and Bowen et al. 2015 (Polecat Bench record). We think it is not very helpful to reiterate the definitions again in the manuscript, citation of those seems to be the best way.

The CIE in the BBCP Polecat Bench drill core has been defined and discussed in Bowen et al. 2015. Assuming, as proposed by the referee above, that the PETM CIE lasted from the onset (118.70 mcd – Bowen et al. 2015) and 55 to 60 mcd (~58.20 mcd) at PCB results in a duration (applying the Table 1 age model) of 157 kyr.

But as written on page 9, line 14: "At the Walvis Ridge ODP sites, the top of the clay layer coincides with the top of the initial rapid recovery of the CIE (Recovery phase I in Murphy et al., 2010). To correlate deep-sea and terrestrial records, the onset and the top of the initial rapid recovery of the CIE are commonly used (McInerney and Wing, 2011)". It is important to know, as written in McInerney and Wing, 2011, that the top of the initial rapid recovery (phase I in Röhl et al. 2007) of the CIE is NOT the top of the subsequent gradual recovery (phase II in Röhl et al. 2007) as assumed by the referee. This matter is complex and can be confusing, but using the definitions as given in Röhl et al. 2007, and more deeply discussed by Murphy et al. 2010, the results of this manuscript clearly show that the duration in the deep-sea is about one precession cycle shorter that in the terrestrial Polecat Bench section.

*In all figures dealing with CIE's correlation between terrestrial and deep-sea records (Figures 7, S12 and S13), the onset of CIE is clearly shown at the abrupt negative δ13C shift, whereas the end of CIE is not obvious neither at ODP sites nor in PCB terrestrial record. It's even sometimes confused when reading the cyclostratigraphic interpretation against the proposed age model, and what is said in the text. For instance, in figure 7, the duration of the entire CIE is assessed at about 180 kyr (120 kyr for the clay layer indicated by the brown rectangle plus 59 kyr till the end of CIE shown by light blue rectangle). In the text, the authors discuss a longer duration of 200 kyr..*

*Again, considering a very likely end of CIE in the terrestrial δ13C data at the top (maximum) of precession cycle no. 8 (Fig. 7), a duration of 168 ky (21 kyr x 8 cycles) could be inferred..*

We admit that the definition of phases needs clarification. The onset of the PETM CIE is pretty clear. "The termination of the CIE at Site 1263 and the ''reference section'' at Site 690 were defined (Tables 1 and 2), by identifying an inflection point in the bulk d13C curve" (Röhl et al. 2007). The inflection point was labeled "G" in Zachos et al. 2005 and used for correlation to other records. It is located at 167.12 mbsf in ODP 690 (Zachos et al. 2005. Table S4). Using the Röhl et al. 2007 age model this point is 153.5 kyr after the onset of the PETM (Table 2 of Röhl et a, 2007; not 171 kyr as written in Murphy et al. 2010)! Using the updated age model developed in this study we obtain an age of 55.749 Ma for inflection point G which translates into 182 kyr between onset and end CIE. As given in Figure 7 of the manuscript.

Looking at Figure 5 of Röhl et al. 2007 the end of the recovery is between cycle 8 and 9 at ODP 690. Based on the cycle counting the duration of the CIE is 8*21=168 or roughly 170 kyr was determined in that paper. In our revision of the age model we simply added one precession cycle, therefore the duration of the PETM sensu Röhl et al. 2007 will be 9*21=189 kyr or roughly 190 kyr. In addition, the position of the onset of the PETM in our manuscript was placed between two precession cycles adding another 7 kyr to the duration (compare the relative age of precession cycle 2: in Röhl et al. this is 24 kyr after the onset, in our study it is 31 kyr after onset). Summing up, the PETM CIE duration is 189+7=196 kyr or roughly 200 kyr in the manuscript.

In a revised manuscript we will add a paragraph clarifying this matter. We will also add some details on the definition of the inflection point G, that is problematic because it is located in the very top of ODP 690B-19H and difficult to identify in other isotope records.

*A focus was also given on the duration of clay-layer interval. The clay layer is characteristic of deep-sea environment. What is the degree of reliability of correlation between terrestrial and deep- sea (using δ13C) data that led to the projection of equivalent clay-layer interval into the terrestrial records? Note that this correlation is crucial for the assessment of duration of the clay layer. Could the authors add uncertainties on their stratigraphic correlation?*

This is discussed on page 9, lines 16 to 34 of the submitted manuscript. The onset of the PETM is clearly correlated by the dramatic shift in carbon isotopes. In marine sediments this is the base of the clay layer. The top of the clay layer, in marine sediments of Walvis Ridge, coincides with the top of the initial rapid recovery of the CIE (Recovery phase I in Murphy et al., 2010). The relatively fast rate of carbon exchange between atmosphere and surface (10's of years) and deep (100's of years) ocean reservoirs requires that the rapid recovery in marine and terrestrial records should be recorded at almost the same time. Using the Röhl et al. 2007 age model as time lag of 25 kyr is apparent between the PCB record and marine data. Assuming that this rapid shift should be nearly synchronous, as written in the ms, we concluded that 25 kyr or about one precession cycle could be missing in the marine records due to the severe dissolution at the onset of the PETM.

*In summary, the authors should state clearly in the manuscript how they define the stratigraphic extent of the entire CIE (especially its end) and the projected clay-layer into the terrestrial records, and accordingly they could compare duration estimates between the two realms.*

We will add a paragraph in the revised version dealing with the above issues.

**2) Comparison with previous age models**

*In the outcrops (Bighorn Basin) in the Polecat Bench section, Abdul-Aziz et al. (2008) arrived to a duration of 157 kyr for the entire CIE-PETM.*

*Westerhold and co-authors cited Abdul-Aziz et al.'s (2008) study, but they did not explain the 157 kyr shorter duration compared to their longer duration of 200 kyr obtained from Polecat Bench drill cores. Given both studies are based on precession cycle counting from the same basin (and the same Polecat Bench site), I strongly recommend that the authors explicitly discuss the source of such significant difference. Although the authors evoked promptly this difference (Page 7, lines 21-23), but it is still ambiguous how they found a longer duration with regard to a shorter duration provided by Abdul-Aziz et al. (2008) (see also 'Comment 1' above).*

*Note that Abdul-Aziz et al.'s (2008) duration estimate (i.e., 157 kyr) is close to the 171 kyr duration of Röhl et al. (2007) inferred from deep-sea records.*

"The main body of the CIE spans ~5.5 precession cycles, or ~115 k.y., and the recovery tail of the CIE spans 2 precession cycles, or ~42 k.y." (157 kyr) – Abdul-Aziz et al. 2008.

Again, as already discussed above, the issue here is the definition of different phases of the PETM. The duration for the main body of the PETM, as written in the ms, is almost identical to Abdul-Aziz et al. 2008. The recovery phases of the PETM CIE have been defined in deep-sea records (Zachos et al. 2005, Röhl et al. 2007). Rapid recovery from the CIE should be nearly synchronous in both records. But it is rather difficult to identify the end of the recovery phase (the inflection point G mentioned above) in other records (including the PCB records) than ODP 690. In Abdul-Aziz et al. 2008 the recovery is from ~63 to ~77m, a distance of 14 m (their Figure 3) containing two precession cycles. The new higher resolution data for the PETM CIE from Bowen et al. 2014 show that the recovery starts (note that the depth in the core is from top down, in the outcrop from bottom up) at 75m and ends at 55m, a distance of 20m. We do not want to discuss here which definition at Polecat Bench is correct, but rather point to the fact that is comes down to this definition to find out the duration of the PETM at Polecat Bench. Concerning the marine records, and applying the definitions given in Zachos et al. 2005 and Röhl et al 2007, the duration of the PETM determined in our manuscript remains at 196 kyr, roughly 200 kyr.

In a revised manuscript, we will clarify this by pointing to the rather difficult identification of the inflection point G in the Polecat Bench records.

**3) Amplitude modulation (AM) of the precession by the eccentricity**

*The authors outlined 'AM of the precession by the eccentricity' in the text body and they also pointed it out in the abstract and conclusions, however, there is no statistical test (or even an attempt by visual inspection) to show or retrieve such modulation. If the authors would still retain this result, then they should demonstrate it, at least at the short eccentricity band.*

*The authors stated (Page 7, lines 8 and 9) : "The filter of the precession cycles of ~8.2 m in both data show modulations that are consistent with eccentricity". Filtering is not sufficient to draw such conclusion. Here a Hilbert transform is required to extract such AM envelopes...*

We will add a Hilbert transform of the data to figure 5 – see below.

[Figure]

**Figure 1** – Modified Figure 5 for the main manuscript: Cyclostratigraphy for Polecat Bench. From left to right: PCB-A(red) and PCB-B (blue) soil nodule carbon isotope data (Bowen et al. 2015), a* data from color scanning, XRF Fe intensity data (in total counts area *1000). Then Gaussian filter of the longer 8.2 m cycle (precession) of the Fe data (lines) and from a* data (dashed lines). Numbers mark the precession cycle counting 5 starting at the PETM, positive numbers is time after PETM, negative numbers before. On the right, Gaussian filter of the 3.5 m cycle (half-precession) and the amplitude modulation of the Fe data extracted by Hilbert transform using the Astrochron software package.

**4) Half-precession**
*Precession vs half-precession ratio is not consistent with the selected bandwidths used for filtering (see for e.g., Fig. 5). Visual inspection in figure 5 indicates that several precession cycles do not match two 'half-precession' cycles, making the hypothesis of 'half-precession' implausible. Also, if the precession central wavelength is 8.2 m, then 'half-precession' central wavelength should be around 4 m (not 3.45 m).*
*Can the authors resolve this mismatch, by changing the bandwidth for example, or abandon the hypothesis of 'half-precession'.*

*In addition, the authors stated (Page 5, Lines 28-29) "The two longer cycles around 8 and 3.5 m have been interpreted as precession and half precession cycles also present in Plio-Pleistocene successions (see Abdul-Aziz et al., 2008)."*

*Abdul-Aziz et al. (2008) did not interpret the 3.5 m cycles as half-precession. Instead, they interpreted them as sub-Milankovitch (or millennial). They even stated in their paper «However, the exact origin of sub-Milankovitch cycles remains enigmatic. ». Sub-Milankovitch (or millennial-scale) cycles do not imply half-precession cycles...*

This was mentioned by referee #1 as well. We will correct this in a resubmitted version of the manuscript. However, it is not important for the cyclostratigraphy which is based on the recognition of the precession cycle only.

**5) Significance of changes in sediment a* color reflectance and Fe content in terrestrial records**

*Although the authors evoked very promptly the potential significance of XRF iron intensity in terrestrial sediments by referring to previous studies (Abels et al., 2012), [and this topic is beyond the scope of the present study], I suggest that the authors develop a little bit the significance of such proxies in terms of climate change (astronomically forced climate). Orbitally driven fluctuations in Fe content in deep-sea sedimentary records have generally (and extensively) been attributed to the relative contribution from carbonate deposition versus detrital-clay inputs. However, the origin of cyclic change in Fe content in terrestrial environments is not well addressed in the litterature...*

It is not the scope of the manuscript to discuss and explore the nature of Fe variations and its direct links to climate change. This requires detailed geochemical analysis, as already done in Kraus et al. 2015 (Palaeogeography, Palaeoclimatology, Palaeoecology 435 (2015) 177–192; http://dx.doi.org/10.1016/j.palaeo.2015.06.021) at Polecat Bench, on thre PCB drill cores. Our focus is on using the apparent cyclicity for age model construction.

The XRF core scanning method applied provides semi-quantitative information of bulk iron concentrations. It does not allow to distinguish oxidation states of Iron necessary to address imprints of climate change on the sediment Fe composition as done in Kraus et al. 2015. Looking at Fe only it is not possible to speculate about humidity, this can be done by combining elemental information into ,e.g., the chemical index of alteration (CIA) done for Polecat Bench by Kraus and Riggins (2007). We are currently working on exactly this topic towards an additional manuscript dealing with XRF core scanning data from the BBCP drill cores. We would like not to include the discussion of the potential significance of XRF iron intensity in the BBCP records because this will be focus of a subsequent manuscript following the our here presented age model study.

**Minor points**

We will correct the revised manuscript as pointed out by the referees recommended edits.

---

## Author Response (AR1)

✉ Universität Bremen **I MARUM I** 28359 Bremen

Dr.
**Thomas Westerhold**
Research Scientist

Leobener Strasse 8
MARUM building, Room 0220
28359 Bremen – Germany

Telefon   +49 421 218 – 65672
E-Mail    twesterhold@marum.de
www      www.marum.de

**To the Editors of**

*Climate of the Past*

.

23/January/2018

Dear Yves Godderis (Editor),

Please accept our submission of the revised manuscript entitled "Synchronizing early Eocene deep-sea and continental records – cyclostratigraphic age models for the Bighorn Basin Coring Project drill cores" for consideration by *Climate of the Past*.

Thank you very much for your editorial work and suggestions. We greatly appreciate your effort and would like to thank the two referees for their critical reviews, all of which improved the manuscript. Comments and constructive criticism encouraged us to carefully revise the manuscript.

Detailed answers to the comments of the referees were already posted on the *Climate of the Past Discussions* web page. Here we address all issues raised by the referees point-by-point to improve the manuscript.

We hope that the revised manuscript now meets the requirements to be published in *Climate of the Past*.

Kind Regards,
Thomas Westerhold, Ursula Röhl, Roy Wilkens, Philip Gingerich, Will Clyde, Scott Wing, Gabe Bowen, Mary Kraus

**The original referees' comments are indicated by italics, our response is marked by red letter.**

**Referee #1**

**MAJOR POINTS**

***Title****: Looking at the ms, one may ask why new is included. Are these new cyclostratigraphic age models for the BBCP cores, but then what are the old cyclostratigraphic age models from the project. Moreover, these age models are also not particularly new when compared with the existing age models based on outcrops, as these are largely identical.*

We change the title of the manuscript to "Synchronizing early Eocene deep-sea and continental records – cyclostratigraphic age models for the Bighorn Basin Coring Project drill cores".

***MTM spectra 1****. The MTM spectra might be somewhat problematical as the null- spectrum and confidence limits do not follow the shape of the spectrum very well. Is this a problem of using the "wrong" model for calculating the null-spectrum (as in Vaughan et al., 2011)? Is it as such logical that the power of all thicknesses between ~ 2 and ~ 20 m plot above the 99% CL? This band contains the dominant 3.5 and 8 m cycles, but constant power above 99% for such a broad frequency is not very logical.*
*In this respect, the authors should preferably not use the Mann and Lees (1996; ML96) robust red noise approach that is in SSA-MTM tookit, as it has a tendency to artificially create low frequency cycles (as documented in Meyers, 2012; example in his Figure 2D illustrates that there is a 90% chance of getting false long period cycles from noise). If they want to use the ML96 approach, they should use the one in Astrochron, which fixes this 'edge-effect' problem.*

All MTM spectra have been replaced by mtmML96 spectra using the Astrochron software package. The methods section was modified accordingly.

*The MTM spectra often show a bewildering numbering of peaks in the frequency band that is of primary interest for the paper. This large number of peaks likely stems from the very long and high-resolution character of the records, but it might be preferable to attempt reducing the number of spectral peaks in this band, as less peaks / resolution imply greater stability of peak position, and is more easy to interpret.*

We agree. Due to the high resolution of the expanded records the spectra show lots of details. Because the spectra have been calculated for the entire length of each record, changes in sedimentation rates will produce several closely spaced peaks. To be able to identify changes in sedimentation rates the evolutive spectra were calculated. For the extraction of cycles by filtering of the signal we applied a 30% bandwidth to compensate for these variations in cycle thickness as described in the manuscript.

***MTM spectra 2****. The dramatic reduction of power at very low frequencies in their MTM spectra is due to the data detrending. My concern is that the frequencies they are interested in are up against this detrended region of the spectrum. So if they want to do this detrending, they should also show power spectra without the trends removed, so one can better evaluate the nature of the peaks (if they are real or a consequence of detrending).*

We now provide MTM spectra using the mtmML96 Astrochron routine for all data. The detrending removed cycles longer than 10m, not relevant for the study.

***Half-precession****. The authors constantly use the term half precession cycle for their 3.5 m cyclicity while their precession related cycles are often more than twice as thick. This problem was also encountered by Abdul-Aziz et al. (2008), when studying the Polecat Bench and Red Butte sections;*

*they concluded that this cycle, which is related to prominent individual paleosols, does not represent half (or semi-) precession, but has a period that is significantly shorter and closer to that of the Heinrich events of the last 100.000 year. This was confirmed by the results of bandpass filtering, and the same is the true for the results of the filtering in the present ms (see their Figure 5 where more than two cycles fit into one precession-related cycle). Hence, the authors should use sub-precession or millennial-scale rather than half-precession.*

We have corrected this in the resubmitted version of the manuscript. However, it is not important for the cyclostratigraphy which is based on the recognition of the precession cycle only.

***Precession minimum*** *(l. 24, p. 9). The authors mention that the PETM onset was in a precession minimum, but it is not clear where that comes from (they refer to see above). Do they mean the Fe-minimum between their cycle 1 and -1? But where is the phase relation with precession based on, or it this a mistake? It is also not perfectly clear to me how that might explain the 4-kyr discrepancy in addition of the one precession cycle misfit, also because discrepancies of 4-kyr are within the uncertainty of all the age models.*

We have clarified this in the revised version of the manuscript. The following was added after (page 7, line 19): "… also assume that the onset of the PETM is located in the minimum between cycle -1 and 1": *Regarding the phase relationship, whether paleosols correspond to precession minima or maxima (Abels et al., 2013) is unknown and not relevant for establishing a cyclostratigraphy based on cycle counting itself. For simplicity we assume that paleosols correspond to more negative bulk carbonate carbon isotope values in more clay rich layers in the deep-sea records (higher Fe XRF intensities, Zachos et al., 2010; Littler et al., 2014; Zeebe et al. ,2017). Thus, the onset of the PETM was set into a precession minimum in the PCB cyclostratigraphy model.*

***Comparison between cyclostratigraphic and Helium-based age models.*** *(l.10-28, p.10). This discussion is becoming a bit semantic and potentially far-fetched, going into much detail, which may not all be that relevant to explain the observed major offset of 40-kyr for the initial rapid recovery of the CIE. Also, the average duration of a precession cycle remains an average and longer precession periods are generally found in intervals with high eccentricity (maxima) and shorter cycles during eccentricity minima (as the different cycles are differently modulated), so this plays a role as well. However, again the difference might be either too small to explain the offset or will only (slightly) enhance it. A relevant question to ask is whether the Helium isotope ratio is not affected (or not) by the enhanced volcanic activity at that time as the East Greenland flood basalts may have formed at the same time (see Wotzlaw et al.).*

We thank the referee for this comment. A sentence is added mentioning the Wotzlaw et al. 2012 study. We think discussing or showing the effects of your new age model compared to the Helium age model is important and could be basis for further research, but we refrain from going into too much detail because this is not the scope of our study.

***Almost****. A very interesting and also intriguing aspect is the potential causal connection between Biohorizon B in the Bighorn Basin and the calcareous nannofossil events of the same age in the marine realm, intriguing especially as the proxies in the marine record do not indicate that something dramatic is happening. However, the authors use the curious wording almost when comparing the continental faunal turnover with the marine events. But almost is not of the same age, so what is exactly the difference and how does almost the same age translate to potentially having the same origin. In Figure 8, the age difference between the two events is ~ 40-kyr, but what are the uncertainties in the respective age models and in the position and thus age of the respective bio-events? The uncertainty in the position of Biohorizon B might be quite large compared with that in the marine record, so do these uncertainties overlap? The reason to develop high-resolution astrochronologic age models is meant to increase the temporal resolution and solve possible*

*temporal relationships and chicken-and- egg problems. So what does this almost imply in this case? This should be made more clear in the ms.*

    This section is rewritten now (as outlined in the discussion phase), the position has been revised and clarified including a proper error discussion and updated Figure 8.

**MINOR POINTS**

***Cycle 0****. Why is there no cycle 0 in the numbering of the cycles in the PCB cores*
    Why should there be a cycle 0? It is not clear what the referee is pointing at. The onset of the PETM was chosen as the zero line (Table 1 of the ms), in-between precession cycle -1 (before onset PETM) and 1 (after onset PETM).

***405-kyr minimum*** *(p.7, l. 14). Is this also not part of a very long 2.0 Myr eccentricity cycle, see Lourens et al., (2005) and Meyers (2015)? Please check what you mean exactly.*
    It is correct as written. The cycles occur in a 405-kyr minimum, a time of low amplitude modulation of the precession cycle by eccentricity.

***3.2 Time series analysis of BBCP drill cores****. The first paragraph before 3.2.1 belongs to the Material & Methods section rather than to Results.*
    We would like to keep this section where it is because the time series analysis and thus age model development should ideally be in one chapter.

**Referee #2**

**MAJOR POINTS**

**1) The definition of the end of CIE-PETM in the Polecat Bench δ13C record**
*Duration estimates of the PETM and comparison with previous studies depend tightly on how the stratigraphic extent of CIE is defined. While it is easier to define the onset of CIE at both realms, the definition of its end is more problematic, especially in the continental δ13C record.*
*In the present form of the ms, it is not clear how the authors (or may be by referring to previous papers) set the end of CIE in the δ13C terrestrial record. Based on their figure 5, I can place it appropriately at 60 mcd depth, and largely at 55 mcd depth. This implies respectively 7.5 to 8 precession cycles, yielding respective durations of 157.5 and 168 kyr (21 kyr mean precession period). These durations are close to the 171 kyr estimate inferred from deep-sea records (Röhl et al., 2007).*

    Definitions for the different phases of the PETM in the deep-sea and the terrestrial realm are given in Zachos et al. 2005, Röhl et al. 2007, Murphy et al. 2010 (all three deep-sea) and Bowen et al. 2015 (Polecat Bench record). We think it is not very helpful to reiterate the definitions again in the manuscript, citation of those seems to be the best way.
    The CIE in the BBCP Polecat Bench drill core has been defined and discussed in Bowen et al. 2015. Assuming, as proposed by the referee above, that the PETM CIE lasted from the onset (118.70 mcd – Bowen et al. 2015) and 55 to 60 mcd (~58.20 mcd) at PCB results in a duration (applying the Table 1 age model) of 157 kyr.
    But as written on page 9, line 14: "At the Walvis Ridge ODP sites, the top of the clay layer coincides with the top of the initial rapid recovery of the CIE (Recovery phase I in Murphy et al., 2010). To correlate deep-sea and terrestrial records, the onset and the top of the initial rapid recovery of the CIE are commonly used (McInerney and Wing, 2011)". It is important to know, as written in McInerney and Wing, 2011, that the top of the initial rapid recovery (phase I in Röhl et al. 2007) of the CIE is NOT the top of the subsequent gradual recovery (phase II in Röhl et al. 2007) as assumed by

the referee. This matter is complex and can be confusing, but using the definitions as given in Röhl et al. 2007, and more deeply discussed by Murphy et al. 2010, the results of this manuscript clearly show that the duration in the deep-sea is about one precession cycle shorter that in the terrestrial Polecat Bench section.

*In all figures dealing with CIE's correlation between terrestrial and deep-sea records (Figures 7, S12 and S13), the onset of CIE is clearly shown at the abrupt negative δ13C shift, whereas the end of CIE is not obvious neither at ODP sites nor in PCB terrestrial record. It's even sometimes confused when reading the cyclostratigraphic interpretation against the proposed age model, and what is said in the text. For instance, in figure 7, the duration of the entire CIE is assessed at about 180 kyr (120 kyr for the clay layer indicated by the brown rectangle plus 59 kyr till the end of CIE shown by light blue rectangle). In the text, the authors discuss a longer duration of 200 kyr..*
*Again, considering a very likely end of CIE in the terrestrial δ13C data at the top (maximum) of precession cycle no. 8 (Fig. 7), a duration of 168 ky (21 kyr x 8 cycles) could be inferred..*
We admit that the definition of phases needs clarification, which is now done in the discussion by adding 17 lines of text to the discussion.

*A focus was also given on the duration of clay-layer interval. The clay layer is characteristic of deep-sea environment. What is the degree of reliability of correlation between terrestrial and deep- sea (using δ13C) data that led to the projection of equivalent clay-layer interval into the terrestrial records? Note that this correlation is crucial for the assessment of duration of the clay layer. Could the authors add uncertainties on their stratigraphic correlation?*
This was discussed on page 9, lines 16 to 34 of the submitted initial manuscript and is still part of the revised manuscript. The onset of the PETM is clearly correlated by the dramatic shift in carbon isotopes. In marine sediments this is the base of the clay layer. The top of the clay layer, in marine sediments of Walvis Ridge, coincides with the top of the initial rapid recovery of the CIE (Recovery phase I in Murphy et al., 2010). The relatively fast rate of carbon exchange between atmosphere and surface (10's of years) and deep (100's of years) ocean reservoirs requires that the rapid recovery in marine and terrestrial records should be recorded at almost the same time. Using the Röhl et al. 2007 age model as time lag of 25 kyr is apparent between the PCB record and marine data. Assuming that this rapid shift should be nearly synchronous, as written in the ms, we concluded that 25 kyr or about one precession cycle could be missing in the marine records due to the severe dissolution at the onset of the PETM.

*In summary, the authors should state clearly in the manuscript how they define the stratigraphic extent of the entire CIE (especially its end) and the projected clay-layer into the terrestrial records, and accordingly they could compare duration estimates between the two realms.*
We have add 17 lines of text to the discussion dealing with this issue.

**2) Comparison with previous age models**
*In the outcrops (Bighorn Basin) in the Polecat Bench section, Abdul-Aziz et al. (2008) arrived to a duration of 157 kyr for the entire CIE-PETM.*
*Westerhold and co-authors cited Abdul-Aziz et al.'s (2008) study, but they did not explain the 157 kyr shorter duration compared to their longer duration of 200 kyr obtained from Polecat Bench drill cores. Given both studies are based on precession cycle counting from the same basin (and the same Polecat Bench site), I strongly recommend that the authors explicitly discuss the source of such significant difference. Although the authors evoked promptly this difference (Page 7, lines 21-23), but it is still ambiguous how they found a longer duration with regard to a shorter duration provided by Abdul-Aziz et al. (2008) (see also 'Comment 1' above).*
*Note that Abdul-Aziz et al.'s (2008) duration estimate (i.e., 157 kyr) is close to the 171 kyr duration of Röhl et al. (2007) inferred from deep-sea records.*

"The main body of the CIE spans ~5.5 precession cycles, or ~115 k.y., and the recovery tail of the CIE spans 2 precession cycles, or ~42 k.y." (157 kyr) – Abdul-Aziz et al. 2008.

Again, as already discussed above, the issue here is the definition of different phases of the PETM. The duration for the main body of the PETM, as written in the ms, is almost identical to Abdul-Aziz et al. 2008. The recovery phases of the PETM CIE have been defined in deep-sea records (Zachos et al. 2005, Röhl et al. 2007). Rapid recovery from the CIE should be nearly synchronous in both records. But it is rather difficult to identify the end of the recovery phase (the inflection point G mentioned in the revised manuscript) in other records (including the PCB records) than ODP 690. In Abdul-Aziz et al. 2008 the recovery is from ~63 to ~77m, a distance of 14 m (their Figure 3) containing two precession cycles. The new higher resolution data for the PETM CIE from Bowen et al. 2014 show that the recovery starts (note that the depth in the core is from top down, in the outcrop from bottom up) at 75m and ends at 55m, a distance of 20m. We do not want to discuss here which definition at Polecat Bench is correct, but rather point to the fact that is comes down to this definition to find out the duration of the PETM at Polecat Bench. Concerning the marine records, and applying the definitions given in Zachos et al. 2005 and Röhl et al 2007, the duration of the PETM determined in our manuscript remains at 196 kyr, roughly 200 kyr.

In a revised manuscript, we have clarified this by pointing to the rather difficult identification of the inflection point G in the Polecat Bench records.

**3) Amplitude modulation (AM) of the precession by the eccentricity**
*The authors outlined 'AM of the precession by the eccentricity' in the text body and they also pointed it out in the abstract and conclusions, however, there is no statistical test (or even an attempt by visual inspection) to show or retrieve such modulation. If the authors would still retain this result, then they should demonstrate it, at least at the short eccentricity band.*
*The authors stated (Page 7, lines 8 and 9) : "The filter of the precession cycles of ~8.2 m in both data show modulations that are consistent with eccentricity". Filtering is not sufficient to draw such conclusion. Here a Hilbert transform is required to extract such AM envelopes...*
We added a Hilbert transform of the data to the revised figure 5.

**4) Half-precession**
*Precession vs half-precession ratio is not consistent with the selected bandwidths used for filtering (see for e.g., Fig. 5). Visual inspection in figure 5 indicates that several precession cycles do not match two 'half-precession' cycles, making the hypothesis of 'half-precession' implausible. Also, if the precession central wavelength is 8.2 m, then 'half-precession' central wavelength should be around 4 m (not 3.45 m).*
*Can the authors resolve this mismatch, by changing the bandwidth for example, or abandon the hypothesis of 'half-precession'.*
*In addition, the authors stated (Page 5, Lines 28-29) "The two longer cycles around 8 and 3.5 m have been interpreted as precession and half precession cycles also present in Plio-Pleistocene successions (see Abdul-Aziz et al., 2008)."*
*Abdul-Aziz et al. (2008) did not interpret the 3.5 m cycles as half-precession. Instead, they interpreted them as sub-Milankovitch (or millennial). They even stated in their paper « However, the exact origin of sub-Milankovitch cycles remains enigmatic. ». Sub-Milankovitch (or millennial-scale) cycles do not imply half-precession cycles...*
This was mentioned by referee #1 as well. We have corrected this in a resubmitted version of the manuscript. It is not important for the cyclostratigraphy which is based on the recognition of the precession cycle only.

**5) Significance of changes in sediment a* color reflectance and Fe content in terrestrial records**

*Although the authors evoked very promptly the potential significance of XRF iron intensity in terrestrial sediments by referring to previous studies (Abels et al., 2012), [and this topic is beyond the scope of the present study], I suggest that the authors develope a little bit the significance of such proxies in terms of climate change (astronomically forced climate). Orbitally driven fluctuations in Fe content in deep-sea sedimentary records have generally (and extensively) been attributed to the relative contribution from carbonate deposition versus detrital-clay inputs. However, the origin of cyclic change in Fe content in terrestrial environments is not well addressed in the literature...*

It is not the scope of the manuscript to discuss and explore the nature of Fe variations and its direct links to climate change. This requires detailed geochemical analysis, as already done in Kraus et al. 2015 (Palaeogeography, Palaeoclimatology, Palaeoecology 435 (2015) 177–192; http://dx.doi.org/10.1016/j.palaeo.2015.06.021) at Polecat Bench, on two PCB drill cores. Our focus is on using the apparent cyclicity for age model construction.

The XRF core scanning method applied provides semi-quantitative information of bulk iron concentrations. It does not distinguish oxidation states of Iron necessary to address imprints of climate change on the sediment Fe composition as done in Kraus et al. 2015. Looking at Fe only it is not possible to speculate about humidity, this can be done by combining elemental information into ,e.g., the chemical index of alteration (CIA) done for Polecat Bench by Kraus and Riggins (2007). We are currently working on exactly this topic towards an additional manuscript dealing with XRF core scanning data from the BBCP drill cores. We would like not to include the discussion of the potential significance of XRF iron intensity in the BBCP records because this will be focus of a subsequent manuscript following the our here presented age model study.

**Minor points**

*Page 1, Lines 19 to 20*:
*"A consistent stratigraphic framework is required to understand the effect of major climate perturbations of the geological past on both marine and terrestrial ecosystems."*
*Should better be:*
*"A consistent chronostratigraphic framework is required to understand the effect of major paleoclimate perturbations on both marine and terrestrial ecosystems."*
corrected

*Page 1, Line 25:*
*"Bighorn Basin Drilling Project (BBCP, Wyoming, USA)"*
*Please change into: "Bighorn Basin Coring Project (BBCP, Wyoming, USA)"*
corrected

*Page 1, Lines 29 to 30 :*
*"The duration of the PETM is estimated at ~200 kyr for the CIE and ~120 kyr for the pelagic clay layer."*
*Should better be: The duration of the PETM is estimated at ~200 kyr for the CIE and ~120 kyr for the associated pelagic clay layer.*
corrected

*Page 2, Lines 7 & 8:*
*"Both have been studied in great detail in both in deep-sea sedimentary and terrestrial successions (Zachos et al., 2005; Abels et al., 2016)."*
*Into: Both have been studied in great detail in deep-sea and terrestrial sedimentary successions (e.g., Zachos et al., 2005; Abels et al., 2016)."*
corrected

*Page 2, Lines 10, 11 & 12:*

*"The hyperthermal events in outcrops and ocean drill cores can be identified by the characteristic negative carbon isotope excursions (CIEs), although these differ in magnitude (McInerney and Wing, 2011; Bowen, 2013)."*
*Magnitude of PETM CIE should also refer to Sluijs and Dickens (2012) (Global Biogeochemical Cycles 26, GB4005).*
added

**Page 2, Lines 12 & 13:**
*"The CIEs are interpreted as massive inputs of δ13C-depleted carbon to the exogenic carbon pool (see Dickens et al., 2011 for discussion)."*
*Into: "The CIEs are interpreted as due to massive inputs of δ13C-depleted carbon to the exogenic carbon pool (see Dickens et al., 2011 for discussion)."*
corrected

**Page 2, Line 21:**
*"... understanding the future of climate on Earth ..."*
*into: "... understanding Earth's future climate ..."*
corrected

**Page 2, Lines 27 & 28:**
*"Deep-sea records have a much lower sedimentation rate (cm/kyr) compared to the terrestrial records (m/kyr), but have been deposited continuously."*
*Into: Deep-sea records have much lower sedimentation rates in the order of cm/kyr compared to the terrestrial records having sedimentation rates in the order of m/kyr...*
corrected

**Page 2, Lines 28 to 30:**
*"Sedimentation at the terrestrial successions very likely was more dynamic due to the different types of deposition (see Bowen et al., 2015)."*
*Into: "Sedimentation in terrestrial environments was very likely more dynamic due to the different types of deposition (see Bowen et al., 2015). "*
corrected

**Page 2, Lines 30 & 31:**
*"To interpret rates of changes and processes before, during and after the events a detailed age model is required."*
*Into: "To interpret rates of changes of geological processes before, during and after the events a detailed age model is required."*
corrected

**Page 2, Lines 34 to 36:**
*"Estimates for the duration of the PETM from deep-sea records are complicated by severe carbonate dissolution, which forms a clay-rich layer at the onset of the event (Röhl et al., 2007)."*
*Into: "Cyclostratigraphic estimates of the duration of the PETM from deep-sea records are hampered by the lack of carbonate-rich sequences, which are characaterized on sites nearby the paleo-CCD with a clay- rich layer at the onset of the event (Röhl et al., 2007), resulting from severe carbonate dissolution (Zachos et al., 2005)."*
corrected

**Page 3, Lines 11 to 15 :**
*"The purpose of this report is to establish high-resolution age models for the BBCP drill cores based on cyclostratigraphy and integrate existing age models from outcrops. Second, these new BBCP drill*

*cores age models will be combined with deep-sea records to synchronize and improve the available astronomical age model for the PETM and Elmo interval."*
*Remove "Second" or change into: "The main purpose of this report is to establish high-resolution age models for the BBCP drill cores based on cyclostratigraphy and integrate existing age models from outcrops. Second, these new BBCP drill cores age models will be combined with deep-sea records to synchronize and improve the available astronomical age model for the PETM and Elmo interval."*
corrected

**Page 3, Line 31:**
*"... the BSN and GMH and PCB sites." Into:*
*"... the BSN, GMH and PCB sites."*
corrected

**Page 4, Lines 12 to 26:**
*All this paragraph deals with isotope data acquisition, which were already presented in Bowen et al. (2015). Thus, such paragraph should be removed or shortened or moved to the supplementary materials.*
corrected, we removed this particular methods section, now citing Bowen et al. (2015).

**Page 5, Line 15:**
*"3.2 Timeseries analysis of BBCP drillcores", please add a hyphen to 'Time-series' and to 'drill-cores' (or a space 'drill core')*
corrected

**Page 8 (Line 34) and page 9 (Line 1):**
*"During the PETM, massive dissolution of carbonates in the deep sea truncated the record (Zachos et al., 2005), complicating the age model constructions."*
*Into: "During the PETM, massive dissolution of carbonates in the deep sea truncated the cyclostratigraphic record (Zachos et al., 2005), complicating the construction of age models."*
corrected

**Page 9, Lines 31 to 34:**
*Adding a precession cycle in deep-sea records to the clay layer is not well argued (see specific comments).*
*It is likely to miss cycles in XRF Ca records because of the clay layer. However, in Fe and Ba XRF data, cycles are well expressed (see ODP Site 690 in Röhl et al., 2007).*
This has been addressed already in the major comments above.

**Page 10, Lines 5 to 7:**
*Charles et al. (2011) found 8.5 precession cycles at BH9/05 Core for the entire CIE, similar to Röhl et al. (2007), who used ODP 1263 data.*
The recovery recorded in the carbon isotope data of TOC at the BH9/05 Core does not show a rapid but a more gradual recovery. And identification of the inflection "G" is not straight forward and almost impossible. Assuming that the rapid recovery ends at 500m in that section, a relatively small change in the carbon isotope recovery gradient, is 6 precession cycles from the onset. This would imply the same duration for the CIE as the clay layer in the deep sea spanning the onset to rapid recovery. However, we would like not to speculate too much due to the uncertain placement of inflection point "G" and /or in the identification of the rapid recovery extend in the BH9/05 Core.

**Page 11, Line 25:**
*"Fe intensities, core images, and color reflectance data were used ...."*
*into: "Sedimentary records of Fe intensities, core images, and color reflectance data were used ...."*
corrected

***Page 19 (Figure 1 caption)***
*"Location map for ODP sites 702, 1260 and 1263 on a 40 Ma paleogeographic reconstruction in Mollweide projection (from http://www.odsn.de)."*
*into: "Location map for BBCP (Bighorn Basin Coring Project, Wyoming, USA), ODP Leg 208 (Sites 1260 and 1263) and ODP Leg 113 (Site 690) on a 56 Ma paleogeographic reconstruction in Mollweide projection (from http://www.odsn.de, Hay et al., 1999)."*
    *should also refer to the original paper of Hay et al. (1999), and not only to the website*
corrected and citation added

***Page 20 (Figure 2 caption):***
*The Wilkens et al.'s (submitted) ms referred in Figure 2 caption, but cited in the reference list as Wilkens et al. (2017); the paper already appeared, so should be 2017.*
corrected and citation added

***Page 23 (Figure 5 caption)***
*Line 4: '(lines)' into '(solid lines)'*
*Line 5: 'at the PETM' into 'at the onset of PETM'*
corrected

***Page 25 (Figure 7 caption)***
*Line 5: "... extracted Gaussian filter of the PCB XRF Fe intensity data"*
*would better be "... extracted precession cycles using a Gaussian filter of the PCB XRF Fe intensity data".*
    *Could the authors please point the end of the CIE directly on δ C data of PCB and deep-sea records?*
Corrected and indicated

[revised manuscript text omitted]